# Single-cell expression and Mendelian randomization analyses identify blood genes associated with lifespan and chronic diseases

Arnaud Chignon[1], Valentin Bon-Baret[1], Marie-Chloé Boulanger[1], Zhonglin Li[1], Deborah Argaud[1], Yohan Bossé [2], Sébastien Thériault[3], Benoit J. Arsenault[4] & Patrick Mathieu [1 ✉]

The human lifespan is a heritable trait, which is intricately linked to the development of disorders. Here, we show that genetic associations for the parental lifespan are enriched in open chromatin of blood cells. By using blood expression quantitative trait loci (eQTL) derived from 31,684 samples, we identified for the lifespan 125 *cis*- and 559 *trans*-regulated expressed genes (eGenes) enriched in adaptive and innate responses. Analysis of blood single-cell expression data showed that eGenes were enriched in dendritic cells (DCs) and the modelling of cell ligand-receptor interactions predicted crosstalk between DCs and a cluster of monocytes with a signature of cytotoxicity. In two-sample Mendelian randomization (MR), we identified 16 blood *cis*-eGenes causally associated with the lifespan. In MR, the majority of *cis*-eGene-disorder association pairs had concordant effects with the lifespan. The present work underlined that the lifespan is linked with the immune response and identifies eGenes associated with the lifespan and disorders.

[1] Laboratory of Cardiovascular Pathobiology, Department of Surgery, Quebec Heart and Lung Institute/Research Center, Laval University, Quebec, QC, Canada. [2] Department of Molecular Medicine, Laval University, Quebec, QC, Canada. [3] Department of Molecular Biology, Medical Biochemistry and Pathology, Laval University, Quebec, QC, Canada. [4] Department of Medicine, Laval University, Quebec, QC, Canada. ✉email: patrick.mathieu@fmed.ulaval.ca

D uration of life or lifespan is a complex trait, which is determined by environmental and genetic factors. The heritability of the human lifespan is estimated at ~15–30%[1]. A recent meta-analysis of genome-wide association data, including more than a million parental lifespans, has identified 12 loci associated with the lifespan[2]. Genome-wide association studies (GWAS) in different complex traits have highlighted that gene variants are enriched in noncoding regions with *cis*-regulatory activity[3]. These genomic regions are enriched in expression quantitative trait loci (eQTL)[4,5]. We hypothesized that a better knowledge of the functional consequences of regulatory variants on gene expression might also provide significant insights into mechanisms of human aging. In addition, using eQTLs to identify genetically regulated expressed genes (eGenes) may give us the opportunity to determine whether the variants are associated with the lifespan or directly causal by using Mendelian randomization (MR) techniques. The main assumption of MR is that the variables that are measured, referred to as instrumental variables, only affect the outcome through the exposure and without confounders. In other words, if the genetic variants only affect the lifespan through their role in modifying gene expression, then we can assess their likelihood of playing a causal role on this outcome. This strategy of considering independent gene variants in an allelic series as instrumental variables is a powerful tool for causal inference. However, there are some challenges. If a genetic variant is associated with the outcome through an alternative mechanism (often referred to as horizontal pleiotropy), it may lead to an inflation of type I error. However, different statistical approaches have been developed to assess the robustness of the association discovered by MR and mitigate false positives. The Cochran's Q test for heterogeneity and the intercept test in Egger regression are routinely used to detect associations where the instrumental variables provide estimates that may violate the main assumption[6]. Also, the identification of outliers with the MR-PRESSO package is another tool to identify instrumental variables that may associate with the outcome through an alternative exposure and to provide corrected estimates[7]. By combining these approaches, we can perform a robust estimate of causation.

The human lifespan is intricately intertwined with the development of diseases. As such, the trajectory of aging is variable throughout a person's lifespan and may be altered by the different risk factors and disorders. We hypothesized that some genetic variants may exhibit antagonist pleiotropy. That is, some variants may provide a survival advantage or reproductive success earlier in life, but predispose to disease later in life. Rather than focusing on the individual alleles, we were curious whether similar genetic pathways were involved in chronic disease and lifespan potential. Gene expression is controlled in tissue-specific dynamic networks, with some genes coordinating the activity of functional modules. By using networks and causal inference, we sought to examine if genetically determined gene expression identified for the lifespan was also involved with the risk of several chronic human diseases. Thus, by using a multi-pronged approach, the objectives of this work on the lifespan were to: assess the tissue enrichment of genetic association data, discover *cis*- and *trans*-regulated eGenes and assess causal associations in MR, identify loci under positive selection and showing antagonist pleiotropy, evaluate the cellular enrichment of eGenes and infer cell communication pathways by using single-cell gene expression data, and use network as a tool to document pathways and assess the link between lifespan-associated eGenes and disorders. In this work, we leveraged 27 GWAS in order to map eGenes associated with the parental lifespan and risk factors/disorders. We found that genetic regions potentially influencing the lifespan were enriched in open chromatin in blood cells, which regulate eGenes

involved in the control of immunity. Analysis of blood single-cell expression data showed that eGenes were enriched in dendritic cells (DCs) and the modelling of cell ligand–receptor interactions predicted crosstalk between eGenes expressed in DCs and a cluster of monocytes with a signature of cytotoxicity. Three loci under positive selection had antagonist pleiotropy with the lifespan. In MR, we identified blood eGenes associated with the lifespan and a long-livedness. Lifespan-associated eGenes were linked to different disorders and causal inference showed that a majority of relationships were concordant (e.g., eGenes negatively associated with the lifespan were positively associated with the risk of disorders).

## Results

**Mapping and annotation of lifespan genetic association data.** Supplementary Fig. 1 presents an overview schematic of the analysis pipeline. The parental lifespan GWAS is enriched in noncoding intergenic and intronic genomic regions (Supplementary Fig. 2). We were interested in understanding what tissues might be enriched in genetic association data for the parental lifespan. We implemented GARFIELD[8], which uses summary statistics data to generate linkage disequilibrium (LD)-corrected annotations based on data from the Roadmap Epigenomics[9] and the Encyclopedia of DNA Elements (ENCODE) projects[10]. Implementation of GARFIELD on summary statistics of genetic association data totaling 1,012,240 parental lifespans[2] identified significant enrichments in CD19$^+$ primary cells (marker of B cells) ($P = 3.41 \times 10^{-7}$) and GM12892 (a B cell line) ($P = 1.59 \times 10^{-5}$) (Fig. 1a and Supplementary Data 1). Considering the strong enrichment of lifespan genetic association data in immune cells, we leveraged summary statistics of blood *cis*-eQTL data derived from 31,684 samples to map genetically *cis*-regulated eGenes[11] associated with the parental lifespan by using the Functional Mapping and Annotation of GWAS (FUMA) tool. In lifespan loci, individual significant single-nucleotide polymorphisms (SNPs) ($P_{GWAS} < 5 \times 10^{-8}$, $r^2 < 0.6$) and SNPs in LD (see Methods) were mapped to blood *cis*-eQTLs. SNP–gene pairs were filtered for multiple testing correction at false discovery rate (FDR) ($P_{FDR} < 0.05$). In total, 4042 SNP–gene pairs ($P_{FDR} < 0.05$) tagging 125 blood *cis*-regulated eGenes were mapped to lifespan genetic association data (Supplementary Data 2). By using EnrichR, we found that blood eGenes were enriched in gene ontology (GO) for T cell receptor (TCR)- ($P = 5.65 \times 10^{-10}$), antigen receptor- ($P = 1.98 \times 10^{-7}$), and interferon γ-mediated signaling pathways ($P = 9.94 \times 10^{-7}$) (Fig. 1b and Supplementary Data 3). Among the blood eGenes, 18 were listed in the database of Online Gene Essentiality[12] (fold enrichment = 1.94, $P = 0.009$, hypergeometric test) (Supplementary Data 4). In the Open Targets database[13], 13 drugs/antibodies targeting 8 eGenes (Supplementary Data 5) are/were evaluated in different phases of development (phases I–IV), whereas 19 eGenes are predicted to be tractable for the development of small molecules (Supplementary Data 6). As *cis*-regulation involves chromatin interactions, we also used FUMA to analyze chromatin contact between lifespan loci and genes by using chromatin conformation capture (Hi-C) data in GM12878 (B cell line). Lifespan loci were mapped to genes within a window region (250 and 50 bp upstream and downstream of the transcription start site (TSS), respectively). There were 56 individual significant SNPs located in 46 genomic regions involved in 205 intra-chromosomal loopings with distant regions. Among the distant regions interacting with the individual significant SNPs, 124 genes were mapped (Supplementary Data 7) and were enriched in GO for nucleosome ($P = 5.44 \times 10^{-10}$) and chromosome ($P = 6.80 \times 10^{-10}$) assemblies (Supplementary Data 8). Overall, 36 genes were mapped by both blood eQTLs and chromatin interactions (Supplementary Data 9). Figure 1c shows zoomed-in circos

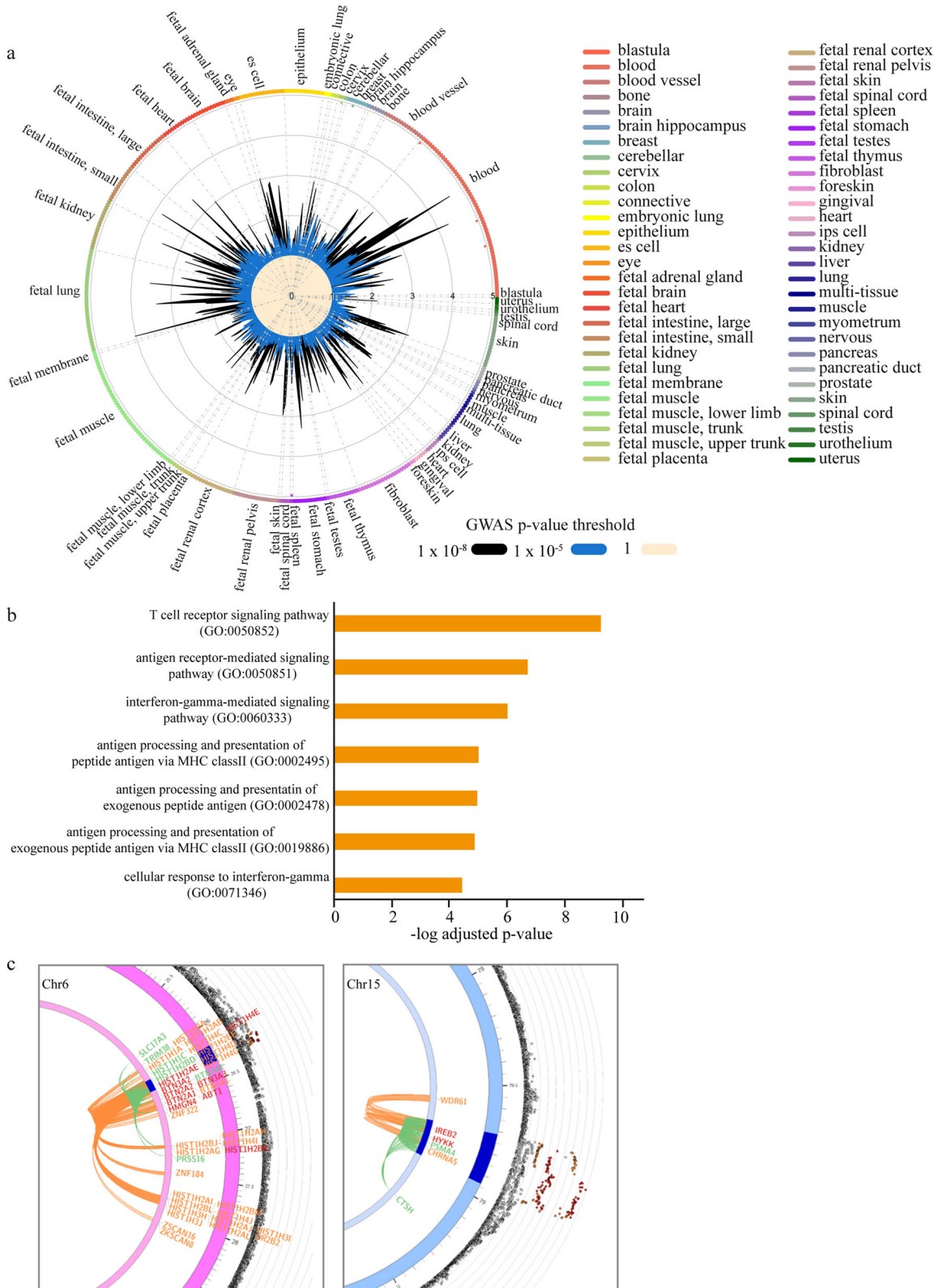

**Fig. 1 Mapping and annotation of parental lifespan GWAS. a** Lifespan gene variant enrichment showed by radial lines (numbers correspond to fold enrichment value) at two GWAS $P$ value thresholds ($P < 1 \times 10^{-5}$ in blue and $P < 1 \times 10^{-8}$ in black) in different tissues and cell types. Significant enrichments are represented by dots in the outer circle. **b** Gene ontology for 125 blood *cis*-regulated eGenes mapped from lifespan; graph showing the seven GO terms with the lowest $P$ values. **c** Zoomed-in circos plots of genetic association data and chromatin interactions in the extended histone locus in chromosome 6 and in the 15q25.1 locus. Names of the genes are in orange for genes mapped only by chromatin interactions, in green for genes mapped only by eQTL, and in red for genes mapped by both chromatin interactions and eQTLs.

plots of genetic association data along with chromatin interactions in the extended histone locus in chromosome 6 and the 15q25.1 locus.

**Network and regulons associated with *cis*-regulated eGenes**. To identify *cis*-regulated eGenes with central functions, we used a network approach by assessing blood-specific co-expression data[14]. We extracted eGene co-expression networks in whole blood from the database of tissue and cancer specific biological networks. This resulted in a network with 4453 nodes and 7321 edges. For visualization purpose, Fig. 2a shows a reduced network topology of key central nodes and interactions. In this network, several eGenes had elevated central betweenness (Supplementary Data 10), which is a measure of nodes with the shortest path acting as bottlenecks between gene modules. The co-expression network in the blood was analyzed for pathway enrichment by using Reactome[15]. The network was enriched in gene expression ($P = 8.08 \times 10^{-23}$), immune system ($P = 1.47 \times 10^{-10}$), and RNA polymerase II transcription ($P = 1.55 \times 10^{-10}$) (Supplementary Data 11). Transcription factors (TFs) and their target genes (regulons) are involved in the control of cell fate[16]. We investigated the regulons of blood eGenes associated with the lifespan by using ChIP-X Enrichment Analysis 3 (ChEA3[17]), which provides TF enrichment for genes by using an expanded list of different sources such as TF–gene co-expression, TF–gene co-occurence, and chromatin immunoprecipitation-sequencing (ChIP-seq) data. TFs were ranked according to their associations with *cis*-regulated eGenes. Most highly ranked TFs were: SP140, interferon regulatory factor (IRF) family and BATF3, an AP-1 member involved in DC differentiation[18] (Supplementary Table 12). A list of blood eGene-derived regulons for SP140, IRF5, IRF8, and BATF3 is provided in Supplementary Data 13. We next wanted to identify immune cells that expressed BATF3 and if they were enriched with lifespan eGenes. Analysis of single-cell RNA-sequencing data of monocytes and DCs (GEO accession number GSE94820) showed that a cluster expressing *CLEC9A*, a marker for a subset of conventional DC (cDC1), was enriched in differentially eGenes including *BATF3*, as well as 21 *cis*-eGenes (fold enrichment = 2.8, $P = 8.74 \times 10^{-5}$, hypergeometric test) (Fig. 2b–d and Supplementary Data 14). Consistently, pathway analysis with Reactome showed that these eGenes were enriched for major histocompatibility complex class II antigen presentation ($P = 6.55 \times 10^{-9}$) (Supplementary Data 15). These data suggest that lifespan eGene-associated regulon may be involved in DC function, including the sensing of dead cell antigens and may thus bridge the innate with the adaptive immune responses.

**MR for the lifespan**. The lifespan GWAS combined fathers and mothers into one parental survival. Effect sizes were reported as the $-\log_e$(Cox hazard ratio), from which years gained or lost could be estimated (see Methods). Causal inference in MR was performed for the blood *cis*-regulated eGenes on the parental lifespan. Independent gene variants ($r^2 < 0.1$) located within ±500 kb from the TSS and associated with the blood expression ($P < 0.001$ corresponds to ~$F$ statistics > 10) were selected as instrumental variables[19] (Supplementary Data 16). The strategy to perform MR analyses is illustrated in Fig. 3a. Enough instrumental variables (minimum 3) to perform MR were available for 116 genes (mean instrumental variables per gene 26) (Supplementary Data 16). After a correction for multiple testing (Bonferroni correction, $P_{causal} < 4.3 \times 10^{-4}$, 0.05/116), 23 eGenes were significantly associated in inverse variance-weighted (IVW) MR with the lifespan (Supplementary Data 17). Among these eGenes, ten did not show heterogeneity on the Cochran's $Q$ test and were considered as lifespan causally associated eGenes (Supplementary Data 17). For the eGenes significant in IVW but with significant heterogeneity, we used the MR-PRESSO package, which tests for the presence of outliers and provides corrected analyses (see Methods). Using this strategy, MR-PRESSO identified six eGenes, for which outliers were detected and provided estimates without distortion after the removal of these outliers (Supplementary Data 18). Hence, by using IVW MR and MR-PRESSO we identified 16 blood eGenes (*PTPN22*, *ARPC3*, *GPN3*, *HECTD4*, *DHX58*, *BECN1*, *CABLES2*, *SLAIN2*, *OCIAD1*, *HIST1H2BF*, *HIST1H4E*, *SH2B3*, *IREB2*, *FES*, *DHX38*, *HSD17B1*), which were considered causally associated with the lifespan (Fig. 3b). Lifespan causally associated eGenes were located on chromosomes 1, 4, 6, 12, 15, 16, 17, and 20 (Fig. 3c). Among these eGenes, only *FES* and *SH2B3* were previously mapped to the lifespan[2]. In model organisms, the deletions of *BECN1* and *OCIAD1* have been previously associated with increased and decreased lifespans, respectively[20,21]. These data including the directional effects are concordant with the present findings in human as the blood expression of *BECN1* and *OCIAD1* were negatively and positively associated with the lifespan, respectively. For *IREB2* ($P_{causal} = 9.56 \times 10^{-23}$), which was also mapped by chromatin interaction (Fig. 1c), an increase of 1 SD in the expression was associated with a gain of 0.75 year (9 months) across the lifespan (see Methods) (Fig. 3b). In the blood co-expression network, a kernel density function analysis showed that lifespan causally associated *cis*-eGenes were distributed along a positive gradient for the degree (hubness) and central betweenness (bottleneck), two metrics indicating prominence in network[22] (Fig. 3d). Of the 16 causally associated eGenes, 10 genes including *IREB2*, *OCIAD1*, *BECN1*, and *PTPN22* were among the top percentile (>99 percentile) nodes with highest betweenness centrality (fold enrichment = 61.7, $P = 1.43 \times 10^{-13}$, hypergeometric test) (Supplementary Data 10). In sensitivity analyses, we tested the 16 blood eGenes in Egger MR, which provides the intercept test as a mean to evaluate horizontal pleiotropy. In Egger MR for the lifespan, we found that *PTPN22*, *DHX58*, *CABLES2*, *SLAIN2*, *OCIAD1*, *IREB2*, *FES*, and *BECN1* remained significant and without horizontal pleiotropy on the intercept test (Supplementary Data 19). The direction of the effect was consistent in IVW and Egger regressions. As an additional measure of sensitivity, we performed Bayesian colocalization analyses between blood *cis*-eQTLs and lifespan genetic association data. This analysis showed that *BECN1* had a strong posterior probability (PP = 0.98) of shared genetic signal between gene expression and the lifespan. For *BECN1*, the colocalization signal is illustrated in Fig. 3e–g by using LocusCompare[23] and shows that rs1011157 is the gene variant with the lowest $P$ values for both *cis*-eQTL and lifespan genetic association data.

**Identification of *trans*-eQTL genes**. We next sought to identify *trans*-regulated genes[11] (>5 Mb from risk loci and/or on a different chromosome) at lifespan causally associated *cis*-regulated blood eGenes loci. The identification of *trans*-regulated genes may provide insights about downstream pathways regulated by *cis*-eGenes. From the individual significant SNPs and variant in LD associated with the lifespan, 4641 SNP–gene pairs ($P_{FDR} < 0.05$) tagging 567 *trans*-regulated blood genes were identified. In total, 559 *trans*-regulated genes were mapped to five lifespan loci, in which causally associated *cis*-regulated eGenes were identified (Supplementary Data 20). Lifespan sentinel variant rs597808 ($P_{GWAS} = 7.32 \times 10^{-13}$), which is a *cis*-eQTL for *SH2B3* ($P_{cis-eQTL} = 7.48 \times 10^{-68}$), was associated with 462 *trans*-eQTL genes. Overall, at *SH2B3* locus, 548 *trans*-eQTL genes were mapped and enriched in GO for cytokine-mediated signaling ($P = 2.66 \times 10^{-9}$), which is consistent with the function of the

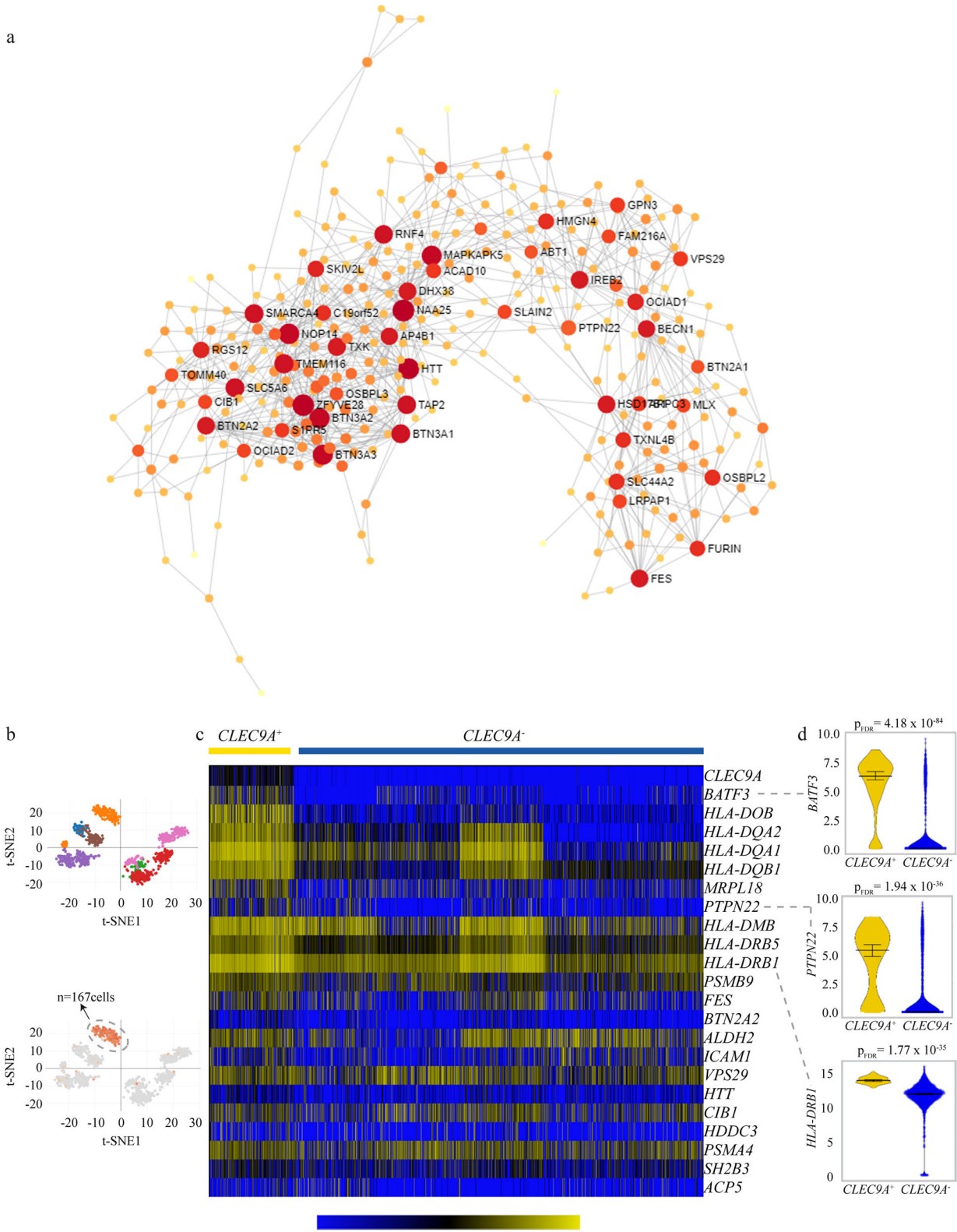

**Fig. 2 Blood eGenes network and cell enrichment. a** Reduced network topology of key central nodes and interactions. **b–d** Analysis of single-cell RNA-sequencing data of monocytes and dendritic cells (GEO accession number GSE94820). **b** t-SNE graphs showing a cell cluster expressing CLEC9A. **c** Heatmap of gene enrichment in CLEC9A− and CLEC9A+ cell clusters. **d** Violin plots showing data distribution (vertical bars illustrate median and 95% CI) and comparing expression in CLEC9A− and CLEC9A+ cell clusters.

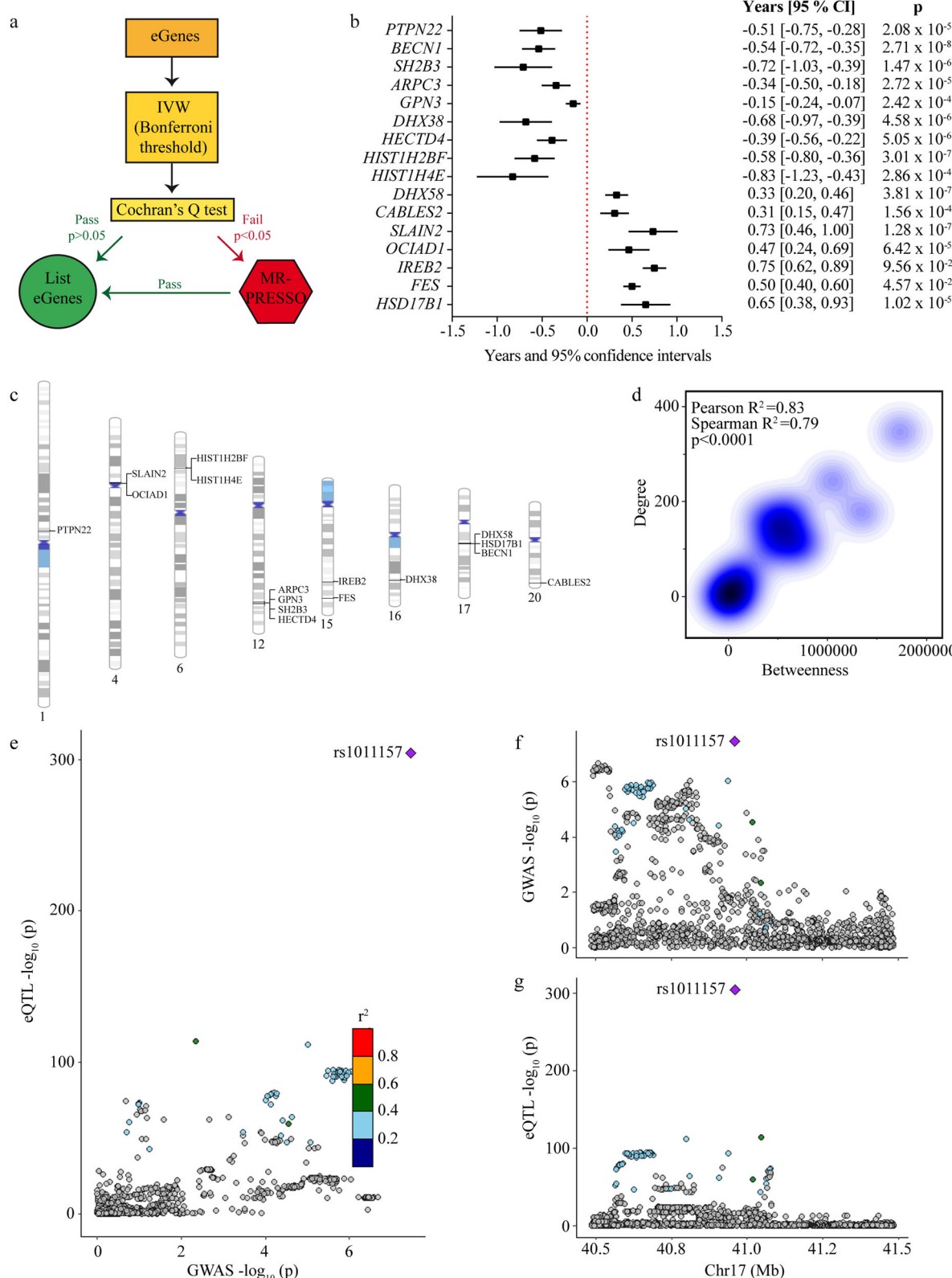

**Fig. 3 Causal inference on lifespan *cis*-eQTL genes. a** Scheme of Mendelian randomization analysis strategy. **b** Forest plots indicating the effects (in years) on lifespan of the 16 blood eGenes considered to be causal. **c** Chromosomal ideograms for the genomic locations of the lifespan causally associated eGenes. **d** Kernel density plots between the degree and the betweenness of the 16 causally associated eGenes, Pearson's and Spearman's correlations. **e–g** Bayesian colocalization analysis between blood *cis*-eQTLs and lifespan genetic association data. **e** LocusCompare showing colocalization signal for BECN1, with rs1011157 having the lowest *P* value for both *cis*-eQTL and lifespan genetic association. **f** Genetic association signal for the GWAS (parental lifespan). **g** Genetic association signal for the blood *cis*-eQTL.

*cis*-regulated eGene *SH2B3*, an adapter molecule for immune signaling[24] (Supplementary Data 21). The *trans*-eQTL genes mapped by SNPs associated with the *cis*-expression of *SH2B3* included 41 TFs curated in TFCheckpoint data[25] for the human (fold enrichment = 1.42, $P = 0.02$, hypergeometric test) and included several regulators of the immune response, such as members of AP-1, STAT, and IRF signaling pathways (Supplementary Data 22). Also, *trans*-eQTL genes associated with the lifespan SNPs at *SH2B3* were enriched in ligand–receptor encoding genes[26] (fold enrichment = 2.27, $P = 1.43 \times 10^{-7}$, hypergeometric test) (Supplementary Data 23). In total, 57 ligand–receptor encoding genes were *trans*-eQTLs associated with the *SH2B3* locus and were enriched in Reactome pathway for cytokine signaling in the immune system ($P = 8.18 \times 10^{-9}$) (Supplementary Data 24). Gene variant rs1230666 (lifespan $P_{GWAS} = 6.44 \times 10^{-9}$) is a *cis*-eQTL for the lifespan causally associated eGene *PTPN22* ($P_{cis\text{-}eQTL} = 1.06 \times 10^{-41}$) and also a *trans*-eQTL for seven genes (Supplementary Data 20). Among these *trans*-eQTL genes, *CD6* ($P_{trans\text{-}eQTL} = 5.17 \times 10^{-7}$), *CTLA4* ($P_{trans\text{-}eQTL} = 5.33 \times 10^{-7}$) and *IL2RA* ($P_{trans\text{-}eQTL} = 2.55 \times 10^{-8}$) are involved in T cell activation, whereas *ARID5B* ($P_{trans\text{-}eQTL} = 3.21 \times 0^{-6}$) is a transcriptional cofactor involved in B cell differentiation[27]. The *trans*-eQTL genes mapped by the lifespan variant rs1230666 were enriched in ARCHS4 for T lymphocyte ($P = 0.001$) (Supplementary Data 25), which is consistent with the high level and function of the *cis*-regulated eGene *PTPN22* in these cells[28–30]. Next, we evaluated if the genetic signal between *cis*- and *trans*-eQTLs was shared by using colocalization analyses. We found strong evidence of shared genetic signal (PP > 0.8) between the region of *cis*-regulated eGene *SH2B3* and *trans*-regulated genes *YWHAH*, *RAB11A*, *PRSS33*, *CLC*, *NCAM1*, *IFI44L*, *RHOB*, and *INPP1* (Fig. 4a and Supplementary Data 26). *PRSS33* is a serine protease with an immune function in eosinophils[31], whereas *IFI44L* is involved in interferon type I response[32]. *NCAM1* (also known as *CD56*) is aberrantly expressed in different malignancies and is a marker of natural killer (NK) cells[33]. Recently, *NCAM1* was found to be expressed by a novel cluster of monocytes (Mono4) with a gene signature of cytotoxicity[34]. Figure 4b shows predicted ligand–receptor interactions derived from single-cell gene expression of peripheral blood mononuclear cells (GEO accession number GSE94820) where monocytes are predicted to interact with DCs through *NCAM1* and different chemokines and immunomodulatory signals for cytotoxicity. Among the ligand–receptor interactions illustrated in Fig. 4b, there are five other lifespan *trans*-eQTL genes associated with the *SH2B3* locus, which are expressed by DCs (*ADRB2*, *TNFRSF14*, *CCR1*, *CD58*) and Mono4 (*KLRB1*) (Supplementary Data 20). Also, we identified a colocalization signal between the *cis*-regulation of *IREB2* and the *trans*-regulation of *PRDM8*, which encodes for a histone methyl-transferase (Fig. 4a and Supplementary Data 26). Finally, the genetic signal for the *cis*-regulation of *FES* was shared with *trans*-regulated genes *SPARC* and *CTTN* (Fig. 4a and Supplementary Data 26). *SPARC* encodes for a cysteine-rich matrix protein involved the control of cell growth[35–37], whereas *CTTN* is involved in the organization of actin[38,39].

**Positive selection at *cis*-regulated eGene loci.** Among the different risk loci associated with the *cis*-regulated eGenes, three overlapped with genomic regions under positive selection in a genome-wide scan (PopHumanScan[40]). Among these loci, two are well-documented regions under selective sweep: the human leukocyte antigen locus in chromosome 6, which has many blood *cis*-eQTLs, is under a balanced selection[41], whereas the *SH2B3* locus has been previously highlighted to be under positive selection[42]. The derived allele for the lifespan sentinel variant rs597808, which

is a blood *cis*-eQTL for *SH2B3* ($P_{cis\text{-}eQTL} = 7.48 \times 10^{-68}$), has an elevated prevalence in population with a European ancestry (frequency in CEU = 0.45) and is associated with a decreased lifespan ($-0.28$ year per allele, $P_{GWAS} = 7.32 \times 10^{-13}$). Gene variant rs597808 is in strong LD with rs3184504 ($r^2 = 0.98$), a coding variant previously associated at genome-wide level with cardio-metabolic traits/disorders[43,44] and autoimmune diseases[45,46]. The other region under positive selection in PopHumanScan is located at 17q21.31 where the lifespan index gene variant rs1011157 ($P_{GWAS} = 3.58 \times 10^{-8}$) (Fig. 5a) is a strong *cis*-eQTL in the blood ($P_{cis\text{-}eQTL} = 5.76 \times 10^{-305}$) for the expression of *BECN1*, a gene involved in autophagy[47]. At this locus, the derived allele T is positively associated with the expression of *BECN1* in the blood and negatively with the lifespan ($-0.33$ year per allele, $P_{GWAS} = 3.58 \times 10^{-8}$). In the blood, an increase of 1 SD in genetically determined expression of *BECN1* was associated with a reduction of 0.5 year (6 months) ($P_{causal} = 2.71 \times 10^{-8}$) across the lifespan. The derived allele at rs1011157 is absent in African populations, whereas it is present in other populations with frequencies varying from 14% in CEU to 35% in Japanese in Tokyo, Japan.

**Lifespan *cis*-regulated eGenes and disorders.** To assess how eGenes related to the lifespan are potentially connected to different disorders, we interrogated DisGenet[48], which provides an expanded and curated database of gene–disease associations. Lifespan blood *cis*-eGenes were significantly enriched in cerebrovascular accident (fold enrichment = 2.16, $P = 0.006$, hypergeometric test), cardiovascular diseases (fold enrichment = 1.91, $P = 0.01$, hypergeometric test) and chronic kidney disease (CKD) (fold enrichment = 2.49, $P = 0.01$, hypergeometric test) (Fig. 6a). We generated a disease network using the DisGenet data, which showed that cerebrovascular accident and cardiovascular diseases were linked to different group disorders, such as lymphoma, hypertensive disease, CKD, autoimmune diseases, and neoplasms (Fig. 6b). We next examined whether blood *cis*-eGenes related to the lifespan were enriched in primary immunodeficiency disorders and cancer. Among the listed primary immunodeficiency disorder genes, we identified five blood *cis*-eGenes (fold enrichment = 3.01, $P = 0.02$, hypergeometric test) (*TAP2*, *CIB1*, *ACP5*, *DCLRE1B*, and *SKIV2L*) that were associated with the lifespan. In PhenoScanner[49], *TAP2*, *DCLRE1B*, and *SKIV2L* have been previously mapped in GWAS to autoimmune disorders such as rheumatoid arthritis (RA) (Supplementary Data 27). Among the genes listed in the COSMIC[50] database for cancer, seven blood lifespan *cis*-eGenes (*ALDH2*, *BCL3*, *CDKN2A*, *FES*, *HIST1H3B*, *SH2B3*, and *SMARCA4*) were identified (fold enrichment = 1.4, $P = 0.22$, hypergeometric test). Thus, these findings suggest that lifespan *cis*-regulated eGenes may be involved in different disease-related outcomes. To assess the role of lifespan causally associated *cis*-eGenes on different risk factors and diseases, we performed MR analyses. In total, causal inference using MR was performed by using summary statistics from 25 GWAS for risk factors/diseases englobing cardiometabolic diseases/traits, autoimmune disorders, atopic disorders, cancer, and neuropsychiatric-behavioral traits/diseases (Supplementary Data 28). Figure 6c illustrates the risk factors/diseases related with the lifespan causally associated *cis*-eGenes. In MR, 117 risk factors/disorders were associated with lifespan causally associated *cis*-eGenes ($P < 0.05$), whereas 28 associations remained significant after a Bonferroni correction ($P < 1.27 \times 10^{-4}$, 0.05/392 traits). Figure 6c provides a color chart for the consistency in the directional effect of the *cis*-regulation on risk factors/disorders and the lifespan. When considering associations remaining significant after the Bonferroni correction, 75% of eGene–disorder association pairs were concordant with the lifespan (e.g., the *cis*-regulated eGene that

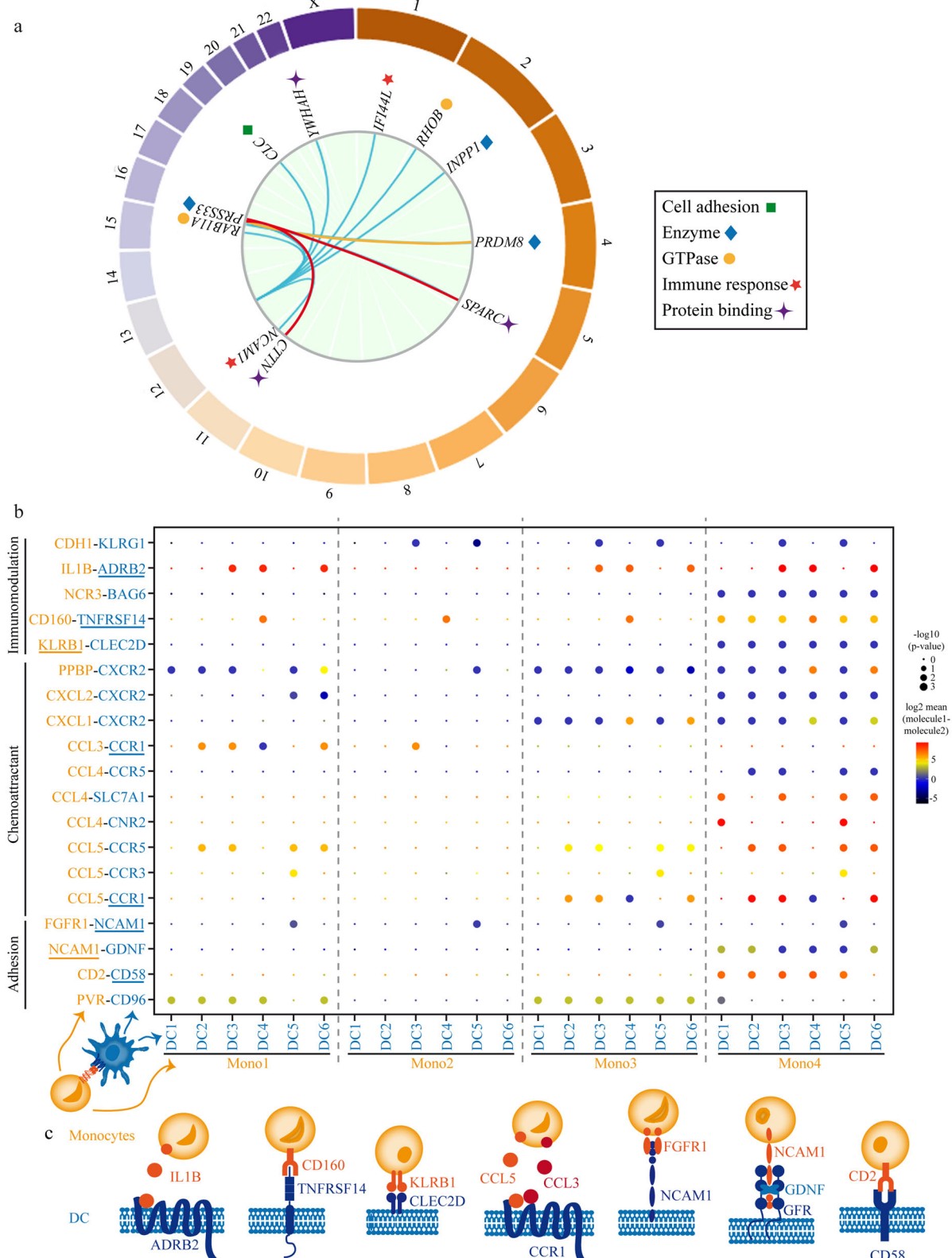

**Fig. 4 *Trans*-eQTLs genes. a** Circos illustrating shared genetic signal between *cis*- and *trans*-regulated genes; *trans*-regulated genes showing shared genetic signal with *cis*-regulated SH2B3 (cyan), IREB2 (yellow), and FES (red). **b** Single-cell gene expression (GEO accession number GSE94820) analysis showing CellPhoneDB predicted ligand–receptor interactions between monocytes and dendritic cells (DCs); on the *y*-axis, molecules identified as lifespan *trans*-QTLs are underlined; molecules expressed by monocytes and DCs are represented in yellow and blue, respectively. **c** Graphic representation of predicted molecular interactions between monocytes and DCs and including lifespan *trans*-eQTLs.

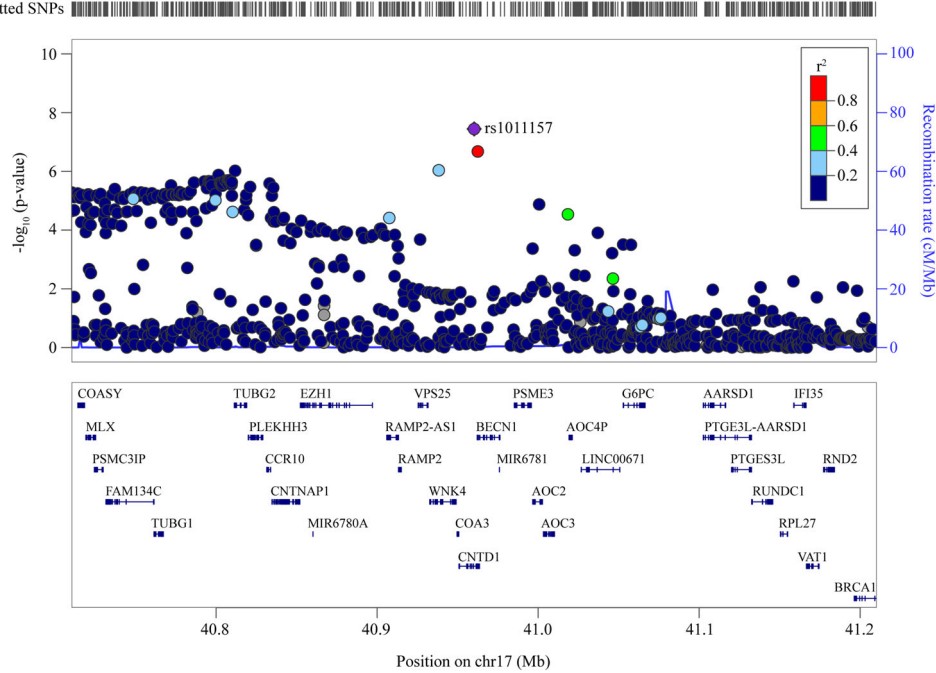

**Fig. 5 Positive selection at *cis*-regulated eGene loci.** Locus zoom of the region surrounding rs1011157, indicated to be under positive selection in PopHumanScan.

decreased the risk of disorder/risk factor increased the lifespan). Some *cis*-eGenes, such as *SH2B3* and *FES*, showed both concordant and non-concordant associations between the lifespan and disorders/risk factors. On the other hand, eGenes such as *BECN1*, *DHX58*, *SLAIN2*, and *PTPN22* were largely concordant in the direction for different risk factor/disease associations with the lifespan. A gain-of-function variant that change an amino acid (R620W) in PTPN22 (allele T at rs2476601) has been previously associated with several autoimmune disorders[51,52]. With regard to the lifespan, gain-of-function variant rs2476601 in *PTPN22* was associated with a reduced lifespan ($-0.27$ year per allele, $P_{GWAS} = 1.63 \times 10^{-5}$). For the expression, the data are consistent with these observations as MR-based causal inference showed that the expression of *PTPN22* in the blood was negatively associated with the lifespan ($-0.51$ year per 1 SD, $P_{causal} = 2.08 \times 10^{-5}$) and positively associated with the risk of RA (per 1 SD odds ratio (OR): 2.62, 95% CI: 2.25–3.06, $P_{causal} = 3.37 \times 10^{-9}$) (Fig. 6c). Other causal associations with the *cis*-expression are in line with previous mapping in GWAS. For instance, genetically determined expression of *FES* was strongly associated with coronary artery disease (per 1 SD OR: 0.90, 95% CI: 0.88–0.92, $P_{causal} = 1.09 \times 10^{-15}$), which is consistent with the genetic signal in GWAS by the tag SNP rs17514846 ($P_{GWAS} = 9.85 \times 10^{-27}$) at the *FURIN/FES* locus. However, other strong relationships in MR, such as the associations between the *cis*-regulation of *OCIAD1* with stroke (per 1 SD OR: 0.87, 95% CI: 0.82–0.91, $P_{causal} = 1.54 \times 10^{-7}$) and breast cancer (per 1 SD OR: 0.78, 95% CI: 0.71–0.86, $P_{causal} = 1.25 \times 10^{-6}$), were not mapped by previous GWAS. These data are in line with a recent study highlighting that a significant proportion of causal associations between *cis*-expression and traits/disorders have no genome-wide significant SNPs in GWAS[53].

**Causal inference for long-livedness.** Longevity as assessed by the parental lifespan does not address whether gene variants and *cis*-regulated eGenes are associated with long-livedness. We thus leveraged and imputed GWAS summary statistics totaling 9793

individuals and examining associations with a long-livedness[54] ($\geq 90$ years) (methods) in order to perform MR analyses. Among the 16 blood *cis*-eGenes causally associated with the lifespan, we could perform MR analyses for long-livedness for 15 genes. In this small series, there were no eGenes significant at a Bonferroni threshold level. However, this analysis showed at an FDR significance level ($P_{FDR} < 0.05$) that genetically determined expressions of *PTPN22* (per 1 SD OR: 0.87, 95% CI: 0.78–0.96, $P_{causal} = 0.04$), *ARPC3* (per 1 SD OR: 0.92, 95% CI: 0.86–0.98, $P_{causal} = 0.04$), *GPN3* (per 1 SD OR: 0.94, 95% CI: 0.89–0.99, $P_{causal} = 0.03$), *HIST1H4E* (per 1 SD OR: 0.85, 95% CI: 0.75–0.97, $P_{causal} = 0.04$), and *HSD17B1* (per 1 SD OR: 1.16, 95% CI: 1.07–1.27, $P_{causal} = 0.01$) were causally associated with a long-livedness ($\geq 90$ years) (Fig. 6d). These association data for long-livedness were consistent with the directional effects of these eGenes on the lifespan.

## Discussion

By using a multi-level approach, which integrated mapping of GWAS, eQTLs, pathway and cell enrichment, MR, and network analyses (Supplementary Fig. 1), we provide evidence that *cis*- and *trans*-regulated blood eGenes are linking the human lifespan with the immune response. Functional annotations of parental lifespan GWAS were enriched in the blood. In total, 16 blood eGenes were causally associated with the lifespan. In a co-expression network, causally associated *cis*-regulated blood eGenes with the lifespan were enriched in nodes with elevated central betweenness, which are shortest path nodes referred to as bottlenecks that link different expressed gene modules. Lifespan-associated eGenes were enriched in cardiovascular disorders, which were linked in a network to CKD, autoimmune disorders, and cancer. In MR, several lifespan-associated eGenes were also associated with the risk of disorders.

Analysis of genetic association data for the lifespan showed a strong enrichment of gene variants in the blood. We found a strong enrichment for TCR- and interferon γ-mediated signaling pathways for the blood eGenes mapped to the lifespan GWAS.

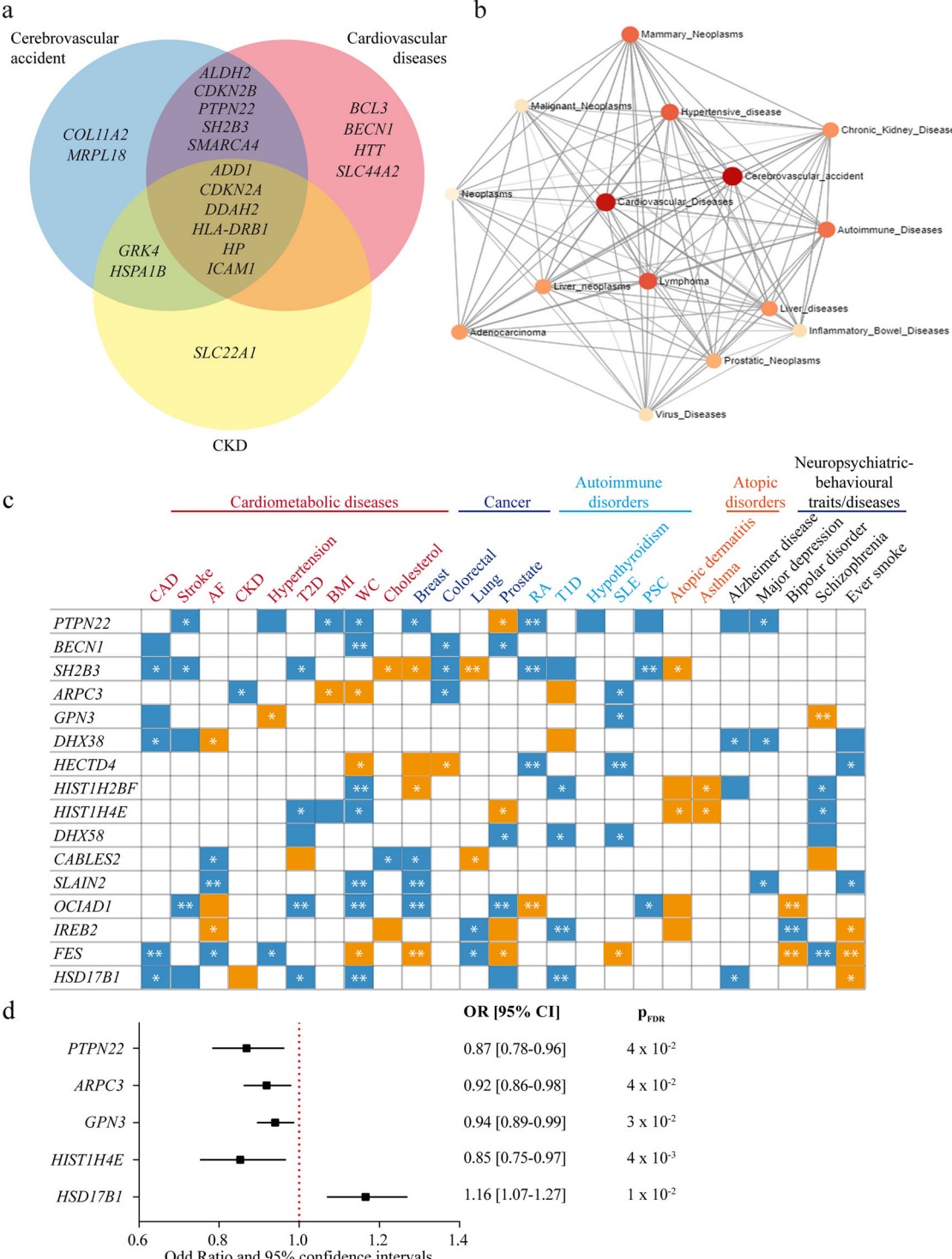

**Fig. 6 Lifespan *cis*-regulated eGenes and disorders. a** Venn diagram of blood *cis*-eGenes associated with cerebrovascular accident, cardiovascular diseases, and chronic kidney diseases (CKDs). **b** Blood *cis*-eGenes disease network showing centrality of cerebrovascular accident and cardiovascular diseases. **c** Chart indicating the consistency in the directional effect of the *cis*-regulation on risk factors/disorders and the lifespan derived from Mendelian randomization analyses; blue: concordant (e.g. the *cis*-regulated eGene that decreased the risk of disorder/risk factor increased the lifespan), orange: non-concordant (e.g. the *cis*-regulated eGene that decreased the risk of disorder/risk factor decreased the lifespan). Blue and orange square $P < 0.05$, *$P < 0.05$ (FDR), **$P < 1.27 \times 10^{-4}$ (Bonferroni). AF: atrial fibrillation, CKD: chronic kidney disease, T2D: type 2 diabetes, BMI: body mass index, WC: waist circumference, RA: rheumatoid arthritis, T1D: type 1 diabetes, SLE: systemic lupus erythematosus, PSC: primary sclerosing cholangitis. **d** Forest plot of genes found to be causally associated with long-livedness (≥90 years).

These data are consistent with the hypothesis that control of inflammation is linked to the aging process[55]. However, the present data suggest that antigen presentation as well as the orchestration of the immune response by T cells may be involved in the human lifespan. We highlighted that lifespan-associated cis-eGenes were enriched in a subset of cDC1 expressing CLEC9A. Studies have underscored that CLEC9A is involved in the recognition and cross-presentation of dead cell antigens[56]. In addition, single-cell expression data revealed that six lifespan trans-eQTL genes (NCAM1, ADRB2, TNFRSF14, CCR1, CD58, and KLRB1) were expressed by DCs and/or a cluster of cytotoxic monocytes, which were predicted to interact. Monocytes/macrophages with a cytotoxic profile have been previously described in tumors[57]. Hence, follow-up studies could provide further mechanistic insights by examining the crosstalk between DCs and monocytes in the context of aging. Autophagy plays a significant role in the function of immune cells, including DCs[58]. We found a strong signal in both colocalization and MR analyses for the expression of BECN1, a key regulator of autophagy, on the lifespan. Recent data in Caenorhabditis elegans (C. elegans) showed that the inhibition of bec-1 (Beclin homolog) in adulthood extended the lifespan[20]. Conversely, in C. elegans the inhibition of bec-1 at the developmental stage decreased the lifespan. Present findings in human showed that lifelong genetically determined BECN1 level in the blood was causally and negatively associated with the lifespan. The expression of IREB2 and DHX58 in blood cells was positively associated with the lifespan. The control of iron metabolism by IREB2 could play a role in host defense against pathogens. DHX58 is a RIG-I-like receptor involved in the interferon γ pathway and host defense against viral infection[59]. Also, some eGenes causally associated with the lifespan, such as FES, SH2B3, and PTPN22, are well known for their role in signaling pathways in immune cells[60–62]. In blood cells, we identified that cis-regulation of SH2B3 was associated with 548 trans-eQTL genes, which were enriched in TFs as well as with ligands–receptors. These data are consistent with the role of the encoded protein as an adapter molecule in the immune signaling cascade. PTPN22 is a protein tyrosine phosphatase highly expressed in lymphocytes and it decreases the signaling mediated by the TCR[63]. Gain-of-function variant in PTPN22 (frequency in CEU = 0.12) was previously identified as being associated to different autoimmune disorders, such as RA, Hashimoto's thyroiditis, Graves' disease, type 1 diabetes (T1D), systemic sclerosis, and systemic lupus erythematosus[64]. Consistently, we underlined that genetically determined expression of PTPN22 was positively associated in MR with the development of RA, thyroid disorder, and primary sclerosing cholangitis. PTPN22 negatively regulates signaling by the TCR and experiments suggest that it may promote the selection of autoreactive clones and affect the function of regulatory T cells[65].

We found that a subset of causally associated eGenes were also associated in MR with long-livedness (≥90 years). The expression of PTPN22 and ARPC3 in blood cells was negatively associated with long-livedness. ARPC3 encodes for a protein that regulates the polymerization of actin and in T cells it participates to the recycling of the TCR[66]. On the other hand, HSD17B1, which encodes for a β-hydroxysteroid dehydrogenase that regulates androgen and estrogen levels[67], was positively associated with an extended lifespan. HSD17B1 is expressed by monocytes and the metabolism of sex steroid hormones could participate to immune senescence[68].

Previous work underlined that genetic association data for cardiovascular diseases showed frequent antagonist pleiotropy with reproductive traits[69]. As such, selective pressure for the fitness operates on traits that occur during the first decades of life. Therefore, it is unlikely that traits and disorders that occur late during a lifetime are under selective pressure. To this effect, in the present study we found that loci identified in a scan for positive selection showed antagonist pleiotropy with the lifespan. At the SH2B3 locus, the derived allele at rs597808 increases the expression of SH2B3 and decreases the lifespan. One study reported that the SH2B3 rs3184504-derived allele, which is in high LD with rs597808 ($r^2 = 0.98$), is associated with an increased production of IL1B through a NOD2 (nucleotide-binding oligomerization domain-containing protein 2) recognition pathway and may have been selected to protect against bacterial infection[42]. Also, the derived allele at rs1011157, which is in a region identified in a genome-wide scan for positive selection, is a strong cis-eQTL for the expression of BECN1 and is negatively associated with the human lifespan. Experimental evidence suggests that BECN1 may participate in host defense against viral infections[70].

The lifespan is a complex trait characterized by a landscape of trajectories, which vary during a lifetime according to complex interactions between the genotype and environmental factors. We found that lifespan-associated blood eGenes were enriched in cardiovascular diseases. In a disease network, we observed that cardiometabolic traits and diseases were linked to autoimmune disorders and cancer. Together, these findings provide support to different observational data. For instance, RA, an autoimmune disorder, is associated with an elevated cardiovascular risk[71]. Hypertension, a cardiovascular risk factor, has been associated with the risk of breast cancer[72]. Lifespan-associated gene OCIAD1 is dysregulated in different cancers[73–75] and recent findings suggest that it controls embryonic stem cell differentiation[76]. In mice, the deletion of OCIAD1 promoted the degradation of p53 along with the development of a myeloproliferative disorder and a reduced lifespan[21]. Consistently, we found that genetically predicted higher expression of OCIAD1 in the blood was associated with an increased lifespan and a reduced risk of breast and prostate cancers. Also, blood cis-regulated eGenes associated with the lifespan were enriched for genes involved in rare primary immunodeficiency disorders. Taken together, these findings highlight that cis-regulation of genes with important functions in immunity and cell fate determination are at the interface of a multimorbid space, which is shared with the lifespan. We found that the majority of the eGene–disorder association pairs significant in MR were concordant for their effects on the lifespan. For instance, positive associations for the expression of eGenes with coronary artery disease, stroke, and dementia were negatively associated with the lifespan. These chronic disorders are well known for their negative associations with the vital prognosis[77,78].

The present work has some limitations. MR is a powerful inference tool; however, only randomized clinical trials can provide a confirmation of causality. The MR analysis performed for long-livedness had limited power. Hence, the associations with long-livedness should be considered exploratory at this stage and could be used as a resource to generate hypotheses for further investigations.

This work provides evidence that genetically determined expression of genes in blood cells is associated with the lifespan. The immune system likely plays a significant role in the trajectory of the human lifespan. Causally associated cis-regulated blood eGenes established connections between a landscape of morbid states and trajectories with the human lifespan and long-livedness. The identification of genetically regulated pathways involved in the lifespan and long-livedness may help develop strategies to provide a healthy aging.

## Methods

**Lifespan genetic associations.** Full summary statistics of genome-wide association data for the lifespan were obtained from 1,012,240 parental lifespans[2], including 691,621 parental lifespans from UK Biobank (excluding SNPs with MAF < 0.005) and 320,619 parental lifespans from LifeGen consortium[79] (excluding SNPs with MAF < 0.01). Lifespan was derived from parental survival (age and alive/dead status) and offspring genotype by using an association test[2]. Parents who

died below the age of 40 years were excluded. The association test was conducted under the Cox's proportional hazards model as described in refs. [79,80]. The Cox's model used a hazard ratio, and to be consistent with the lifespan (which implies a positive effect size for a longer life), the effect sizes were expressed as $-\log_e$(Cox's hazard ratio) corresponding to a $\log_e$(protection ratio). Years of life were estimated as $\log_e$(protection ratio) $\times 10$ according to the average effect across cohorts of the hazard ratio on the lifespan[79]. GWAS summary statistics for the lifespan were publicly available[2].

**Annotation of lifespan genetic associations.** GARFIELD[8] was used to characterize functional, cellular and regulatory contribution of genetic variations for the lifespan. It provides enrichment of genome-wide summary association statistic, which is corrected for the LD structure, in tissue-specific functional elements. GARFIELD uses annotations from ENCODE[10] and Roadmap[9] epigenomics data (1005 features, including genomic annotations, chromatin states, histone modifications, DNaseI hypersensitive sites, and TF binding sites, in a number of cell lines and tissues). The software includes a C++ code for data pre-processing and a R code for fold enrichment, significant testing, and visualization. LD data are included as well as annotation data. $P$ values for each SNP were extracted from the lifespan genome-wide association data summary statistics and default settings were used.

**Mapping of lifespan genetic associations.** GWAS for the lifespan was mapped to blood *cis*- and *trans*-regulated genes by using FUMA[81]. With the FUMA SNP2GENE function, blood *cis*- and *trans*-eQTLs data from 31,684 blood samples[11] (data from https://www.eqtlgen.org/index.html) were used to map genetic associations for the lifespan. Genomic loci associated with the lifespan were defined using a pre-calculated LD structure of the 1000G EUR reference population. SNPs in genomic loci with LD $r^2 < 0.6$, $P$ value $< 5 \times 10^{-8}$ and MAF $\geq 0.01$ were identified as independent significant SNPs (IndSigSNPs). SNPs that have LD $r^2 \geq 0.6$ and MAF $\geq 0.01$ with one of the IndSigSNPs were included as candidate SNPs. These SNPs might not be available in the GWAS dataset, but are available in the 1000G EUR reference population. IndSigSNPs independent from each other (LD $r^2 < 0.1$) were identified as lead SNPs. Genomic loci closely located ($<250$ kb based on the most right and left SNPs of each locus) were merged into one genomic locus. Gene annotation was based on Ensembl (build 85) and entrez ID yielding identification of 19,436 protein coding genes. Blood *cis*-eQTL mapping mapped IndSigSNPs to genes up to 1 Mb apart (called *cis*-regulated blood eGenes), and blood *trans*-eQTL mapping mapped IndSigSNPs to genes $>5$ Mb from genomic loci and/or on a different chromosome. Only significant SNP–gene pairs ($P_{FDR} < 0.05$) were kept.

Chromatin interaction mapping for the lifespan was performed with the FUMA SNP2GENE function using Hi-C data in GM12878 (GEO accession number GSE87112)[82]. IndSigSNPs located into significantly interacting regions were kept and then mapped to genes whose promoter regions (250 bp upstream and 500 bp downstream of the TSS) were located within other significantly interaction regions. Those SNP–gene loops formed frequently interacting regions has described previously[82]. Only significant SNP–gene contacts were kept ($P_{FDR} < 1 \times 10^{-6}$). Circos zoom of genetic associations (eQTL and chromatin interaction) were generated with the integrated tool in FUMA.

**Gene essentiality.** Gene essentiality was defined by using the Online GEne Essentiality database[12] (OGEE), which provides a list of human essential genes based on experimental data. The *Homo sapiens* dataset of essential genes was downloaded to identify essential eGenes. A hypergeometric test was performed to test overrepresentation of essentiality for the eGenes.

**Drug-target identification.** The Open Targets database[13] provides drug-target identification and prioritization based on human genetic data and clinical information. The Open Targets database was used to identify small molecules and/or antibodies targeting the eGenes, their clinical trial status, and tractability for the development of small molecules.

**Network analyses.** A co-expression network analysis was performed to identify eGenes with central functions. NetworkAnalyst[83] was used to generate a tissue-specific co-expression network of eGenes based on whole-blood expression data from the TCSBN database[14]. Pathway enrichment for the network was evaluated by using the Reactome database[15]. Metrics for centrality (degree and betweenness) were obtained from NetworkAnalyst. High betweenness centrality enrichment for the causal eGenes was evaluated by testing these genes against the top percentile (>99 percentile) nodes with the highest betweenness centrality with a hypergeometric test. The Minimum Network tool and the Graphopt layout were used to generate the visual representation of the network. The Origin software was used to make the kernel density plot.

DisGenet[48] data were downloaded to assess the association of eGenes with disorders by using BeFree gene–disease associations, which provides a curated list from text mining in MEDLINE. eGenes associated with diseases available in the BeFree database were integrated in a gene–disease matrix. One gene connected with two diseases formed a disease–disease pair used to generate a edge list. Disease–disease pairs were thus obtained for each eGene and used to perform a disease network for the lifespan using NetworkAnalyst[83] as a visualization tool.

**TFs and regulons analysis.** ChEA3[17] was used to identify TFs associated with the expression of eGenes. ChEA3 provides TFs enrichment for genes based on ChIP-seq experiments (from ENCODE[10], ReMap[84], and data publicly available), co-expression data between TFs and genes (based on processed RNA-seq from GTEx[85] and ARCHS4[86]), and TF-gen[87].

**Single-cell analyses.** Single-cell RNA-sequencing data of monocytes and DCs (GEO accession number GSE94820) were analyzed by using Automated Single-cell Analysis Pipeline (ASAP[88]). Data were filtered for the variance (threshold 50%), log-transformed, clustered by $K$-means, and reduction of dimension with t-SNE at perplexity 30. Differential expression was performed by using Limma[89] at $P_{FDR} < 0.05$. For heatmap representation, log-transformed data were used for $Z$-score distribution and visualized by using Morpheus.

To infer ligand–receptor interactions between monocytes and DCs, the publicly available CellPhoneDB v2.0 package[90] was used. Single-cell RNA-sequencing data of monocytes and DCs (GEO accession number GSE94820) was $\log_2$ transformed and used as an input in CellPhoneDB using default parameters. Significant predicted ligand–receptor interactions were represented by using CellPhoneDB.

**Mendelian randomization.** Causal inference for eGenes on the lifespan was evaluated with two-sample MR by selecting independent SNPs (instrumental variables) associated with the expression of these eGenes. SNPs were analyzed within a window of 500 kb around the TSS of each eGene, then only SNPs strongly associated with the eGene expression ($P < 0.001$ corresponds to ~$F$ statistics > 10) and independent ($r^2 < 0.1$ based on the 1000G EUR reference panel) were selected as instrumental variables. For eQTLGen data, $\beta$ (effect size) and SE were estimated for each instrumental variable from their $Z$-score ($Z$), allele frequency ($p$), and sample size ($n$) (data available from www.eqtlgen.org/cis-eqtls.html) using the following equation[91]:

$$\hat{\beta} = Z \times \widehat{SE},$$

where $\widehat{SE} = 1/\sqrt{2p(1-p)(n+Z^2)}$.

MR was performed by using the MR package[92]. Horizontal pleiotropy was estimated by using the heterogeneity test evaluated with the Cochran's $Q$ test and was considered significant when $P_{heterogeneity} < 0.05$. The MR-PRESSO package[7] was performed for eGenes significant in IVW, but with significant heterogeneity on the Cochran's $Q$ test ($P_{heterogeneity} < 0.05$). MR-PRESSO performs an IVW MR and includes three components: detection of heterogeneity with the MR-PRESSO Global test, correction for heterogeneity via outlier removal with the MR-PRESSO Outlier test, and test of significant difference in the causal estimate before and after removal of outliers with the MR-PRESSO Distortion test. MR-PRESSO was performed if $P_{Global\ test} < 0.05$ and $P_{outliers\ test} < 0.05$; the test was considered significant if $P_{IVW-corrected} < 0.05$ and $P_{Distortion\ test} > 0.05$. Egger regression and the intercept test to evaluate horizontal pleiotropy of instrumental variables were performed as sensitivity analyses. An absence of horizontal pleiotropy was considered if $P_{Intercept} > 0.05$.

**Genetic colocalization analysis.** Shared genetic signals between the expression of *cis*-regulated eGenes (*cis*-eQTLs) and lifespan genetic associations or the expression of blood *trans*-eQTL genes were evaluated using the HyPrColoc package[93]. HyPrColoc provides a Bayesian colocalization analysis across traits in a genomic region in order to test for shared genetic signal. Genomic regions were defined as 500 kb downstream and 500 kb upstream of the TSS of each eGene. As described previously, $\beta$ an SE for each SNPs associated with the expression of eGenes were estimated from $Z$-score, allele frequency, and sample size. A shared genetic signal between the expression of a *cis*-eGene and the lifespan or the expression of a blood *trans*-eQTL gene was considered if the PP was >0.8. LocusCompare[23] was used to visualize the shared genetic signal between a *cis*-eGene and lifespan genetic associations at a locus. Biocircos[94] was used to present colocalization signals between *cis*-eQTL and *trans*-eQTL genes.

**GO, pathway, and enrichment analyses.** GO and pathway enrichment were performed by using EnrichR[87] and data were reported by using adjusted $P$ values. The TFCheckpoint[25] database, which provides experimental evidence of human TFs on gene transcription regulation, was downloaded. The primary immunodeficiency deficiency and ligand–receptor gene lists were downloaded from https://esid.org/Working-Parties/Registry-Working-Party/ESID-Registry/List-of-diseases-and-genes and https://www.nature.com/articles/ncomms8866, respectively[26]. The enrichment for eGenes was performed by using hypergeometric test.

**Positive selection at *cis*-regulated eGene loci.** The PopHumanScan[40] catalog, which regroups regions of the human genome showing strong evidences of positive selection along the human lineage was downloaded. Lifespan loci genomic coordinates were intersected with the positive selection scan from PopHumanScan. Ancestral and derived alleles were identified from the Ensembl 92 VCF file (ftp://ftp.ensembl.org/pub/release-92/variation/vcf/homo_sapiens/homo_sapiens.vcf.gz). Variant allele frequencies were visualized by using The Geography of Genetic Variants Browser[95] (GGV) and data from the 1000 Genomes.

# ARTICLE

**Gene–phenotype associations**. The PhenoScanner[49] catalog containing human genotype–phenotype associations was used to identify immune disorders associated with eGenes enriched in primary immunodeficiency disorders.

**Imputation from summary statistics**. The RAISS[96] package was used to impute summary statistics. It relies on a Gaussian imputation of summary-level data and empirical evidence indicates no increase in false-positive rate compared to imputation on individual data. First, LD-correlation matrix from the 1000G EUR reference population was generated with plink, and $Z$-scores were calculated from long-livedness GWAS summary statistics[54]. RAISS was then run with default settings. $\hat{\beta}$ and $\widehat{SE}$ were estimated from $Z$-scores as described above to perform MR.

**Statistics**. Hypergeometric tests were performed by using the hypergea R package. Spearman's and Pearson's correlations were performed with GraphPad Prism 5.0 (GraphPad Software, Inc., San Diego, CA).

**Reporting summary**. Further information on research design is available in the Nature Research Reporting Summary linked to this article.

## Data availability

We performed analyses based on publicly available GWAS summary statistics (Supplementary Information). As all analyses were based on publicly available GWAS summary statistics, no ethical approval was required. GWAS summary statistics can be found at the links below for the indicated phenotypes.

Lifespan: https://datashare.is.ed.ac.uk/handle/10283/3209; CAD: https://data.mendeley.com/datasets/gbbsrpx6bs/1; stroke: http://www.megastroke.org/download.html; AF: http://csg.sph.umich.edu/willer/public/afib2018/; CKD: https://ckdgen.imbi.uni-freiburg.de/; T2D: http://diagram-consortium.org/downloads.html; BMI: https://portals.broadinstitute.org/collaboration/giant/index.php/GIANT_consortium_data_files; cholesterol: https://www.understandingsociety.ac.uk/; breast cancer: http://bcac.ccge.medschl.cam.ac.uk/bcacdata/icogs-complete-summary-results/; colorectal cancer: https://grasp.nhlbi.nih.gov/FullResults.aspx; prostate cancer: http://practical.icr.ac.uk/blog/; RA: http://plaza.umin.ac.jp/~yokada/datasource/software.htm; PSC: https://www.ipscsg.org/; atopic dermatitis: ftp://ftp.ebi.ac.uk/pub/databases/gwas/summary_statistics/PaternosterL_26482879_GCST003184; asthma: https://genepi.qimr.edu.au/staff/manuelf/gwas_results/main.html; Alzheimer: https://ctg.cncr.nl/software/summary_statistics; major depresssion: https://www.med.unc.edu/pgc/shared-methods/data-access-portal/; bipolar disorder: https://www.med.unc.edu/pgc/data-index/; schizophrenia: https://www.med.unc.edu/pgc/data-index/; ever smoke: https://www.thessgac.org/data; hypertension, waist circumference, lung cancer, T1D, hypothyroidism and SLE: http://www.nealelab.is/uk-biobank; long-livedness: https://grasp.nhlbi.nih.gov/FullResults.aspx.

## Code availability

All software used in this analysis is publicly available at the URLs below:

GARFIELD package: https://www.ebi.ac.uk/birney-srv/GARFIELD/; FUMA: https://fuma.ctglab.nl/; eQTLGen: https://www.eqtlgen.org/index.html; Hypergea package: https://cran.r-project.org/web/packages/hypergea/index.html; Enrichr: https://amp.pharm.mssm.edu/Enrichr/; OGEE: http://ogee.medgenius.info/browse/; Open Targets: https://www.opentargets.org/; NetworkAnalyst: https://www.networkanalyst.ca/; Reactome: https://reactome.org/; ChEA3: https://amp.pharm.mssm.edu/chea3/; ASAP: https://asap.epfl.ch/; Limma: http://bioinf.wehi.edu.au/limma/; Morpheus: https://software.broadinstitute.org/morpheus/; Mendelian randomization package: https://cran.r-project.org/web/packages/MendelianRandomization/index.html; MR-PRESSO package: https://github.com/rondolab/MR-PRESSO; HyPrColoc: https://github.com/jrs95/hyprcoloc; TFcheckpoint: http://www.tfcheckpoint.org/; PopHumanScan: https://pophumanscan.uab.cat/; DisGeNET: https://www.disgenet.org/home/; COSMIC: https://cancer.sanger.ac.uk/cosmic; CellPhoneDB: https://github.com/Teichlab/cellphonedb; PhenoScanner: http://www.phenoscanner.medschl.cam.ac.uk/; RAISS package: https://gitlab.pasteur.fr/statistical-genetics/raiss; UK Biobank: http://www.nealelab.is/uk-biobank.

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

## Acknowledgements

This work was supported by the Canadian Institutes of Health Research grants to P.M. (FRN148778, FRN159697) and the Quebec Heart and Lung Institute Fund. Y.B. holds a Canada Research Chair in Genomics of Heart and Lung Diseases. S.T. and B.J.A. hold a junior scholarship from Fonds de Recherche du Québec-Santé (FRQS). P.M. holds a Fonds de Recherche du Québec-Santé (FRQS) Research Chair.

## Author contributions

A.C. and P.M. designed the study. A.C., V.B.-B., Z.L. conducted analyses of mapping, MR, and colocalization. Z.L. and P.M. conducted analyses for positive selection. Network analyses were performed by P.M., Z.L., and A.C. M.-C.B. performed graphs and integrative figure. Y.B., S.T., D.A., and B.J.A. provided important intellectual inputs. P.M. and A.C. drafted the manuscript. All the authors critically reviewed the manuscript and provided scientific inputs.

## Competing interests

The authors declare no competing interests.
