## [Peer Review File · Communications Biology]

Reviewers' comments:

Reviewer #1 (Remarks to the Author):

In the present work, the authors have conducted a study showing that genetic regions with a potential influence on lifespan are enriched in open chromatin in blood cells regulating a set of expressed genes involved in immune responses. Using a two-sample Mendelian randomization they then identified blood eGenes associated with the lifespan and a longevity. Overall the present work conveys a rather important message that genetically-determined expression of genes in blood cells is associated with the lifespan and that immune responses play a significant role in the human lifespan trajectory. There are, however, certain concerns that need to be clarified or re-addressed.

1. The work rather fails to answer the very question it poses already in the abstract "How the expression of genes (eGenes) connects the lifespan with the risk of disorders". The work provides no mechanism or mechanistic ques but correlations and associations. In particular, the work identified an over-representing gene network related to innate immune signaling in blood cells and then moves on making various, occasionally unnecessary, extrapolations. It also remains unclear what the specificity of the immune-related genes is to other tissues.
2. There have been many and important works connecting DNA damage, inflammation and ageing or immune related responses and longevity. It is not clear in what sense the present work is novel or what the new message is that was unknown prior to this work. This should be clearly stated.

Reviewer #2 (Remarks to the Author):

Mathieu and colleagues applied approaches combined Mendelian randomization, eGenes mapping, functional annotation and network analysis and linked lifespan with the immune response and connected lifespan blood eGenes with chronic disorders. Overall, this is a novel and well-designed work. The findings from this work is important. I have a few concerns.

1. Mendelian randomization used SNPs as instrumental variables for specific exposures. By the Mendelian laws, alleles of SNPs segregate and are randomly inherited from parents to offspring. When using genes for MR analysis, one should be cautious to make causal inferences since multiple SNPs in a single gene may have different function and may not all independent of the outcome.
2. When considering multiple testing, why use 0.05 as cut-off for FDR? How many genes were tested in the model?
3. When evaluating the associations for disorders, it will be interesting to look at the tissue-specific eQTLs and eGenes.
4. Limitations of the study should be addressed.

Reviewer #3 (Remarks to the Author):

Summary

Cignon et al present a study in which the causal function of longevity associated SNPs is explored using a wide range of existing data. This is an excellent case study on the importance of large public datasets and databases. In this work they discovered a number of genes causally associated (by

Mendelian Randomization) with lifespan and long livedness. These causally associated genes are enriched in immune system functions, and their associated pathways are often associated with chronic morbidity. This synthesis raises many interesting questions regarding the role of the immune system in normal aging.

Clarifications & required changes

General

1. The paper is quite dense. Reviewing the paper took much longer than anticipated because of the number of times I had to re-read to get a sense of the overall work. Throughout the entire paper, it would benefit you to more explicitly connect ideas and goals. Many paragraphs would benefit from the use of topic sentences to orient the reader to what will be discussed.

There are also areas where the paper is jargon-dense. Many (of these ideas can be explained in ways that will be accessible to readers that aren't experts in the specific domain.

2. eGenes is variably defined as eQTL genes or expression genes.

Introduction

3. "The strongest association... is in the APOE locus..." An underpinning of the study is that we need to look at eQTLs in non-coding RNA to understand lifespan. Of the 12 loci, how many are actually concentrated in non-coding regions? If of 12 loci the lead SNPs are in non-coding areas in 11/12, this makes good sense. If most are in coding regions, it might not.

Results

4. "Summary statistics of genetic association data..." did you use the total summary statistics, or the summary statistics for the 12 significant loci?

This is also an example of a paragraph that would benefit from a topic sentence. "We were interested in understanding what tissues the lifespan associated loci might be enriched in. One tool to determine these enrichments is GARFIELD, which uses GWAS summary statistics to generate linkage disequilibrium corrected annotations based on data from the Roadmap Epigenomics and Encyclopedia of DNA Elements (ENCODE) projects. GARFIELD analysis of the lifespan GWAS summary statistics found significant open chromatin enrichments..."

5. "highest enrichment in open chromatin detected in CD19+ primary cells... and GM12892". Not all readers are going to be familiar with the GM12892 cell line or what CD19+ cells are. It is probably worth following this up by mentioning what they are.

6. "Considering the strong enrichment of lifespan genetic association data with blood..." I might suggest altering blood to something more like immune cells, PBMCs, B cell, lymphocytes, etc, as the signal is enriched in blood, but particularly in lymphocytes.

7. "...we leveraged summary statistics of blood cis-eQTL data derived from 31,684 samples to map genetic cis-regulated eGenes associated with parental lifespan." to "...associated with parental lifespan using the FUMA tool." Or something similar.

8. "As cis-regulation involves chromatin interactions, we analyzed" to something like "...interactions, we also used FUMA to analyze..." I know you describe these things in the methods section. But the use of so many different tools and DBs makes it hard to follow in the results. A little more detail here and there will help the flow of the paper.

9. "Blood eGenes were enriched in gene ontology (GO) for T cell..." Were these enrichments from GARFIELD, FUMA, or EnrichR?

10. "...using chromatin confirmation capture (Hi-C) data in GM12872." Is this GM12872 or GM12878?

11. "Blood eGenes used as seeds generated..." to something like "We extracted eGene co-expression networks in whole blood from the database of tissue and cancer specific biological networks. This resulted in a co-expression network with..." Unless I've missed how TCSBN works, these aren't seeds so much as simple retrieval requests for annotated networks.

12. "...several eGenes had elevated central betweenness..." The table doesn't seem to be sorted by betweenness or degree. So how are you defining elevated? Greater than the average for the network? The top n %? Percentile? Is there an obvious cutoff?

13. "...A list of blood eGene-derived... provided in Suppl. Table 13." This doesn't have to be changed, but this is an included column in ST12.

14. "Analysis of single-cell RNA sequencing data of monocytes..." Again, no transition. Fill the reader in on your logic. "We then wanted to identify what blood cells most expressed BATF3. Analysis of single cell RNA sequencing data..."

15. Suggest moving the whole part about "... involved in cross-presentation of necrotic..." to somewhere with the "These data suggest that lifespan..." It's a bit out of place where it is there. If you put it after the DC association you could then give that brief tidbit about what the cDCs could be doing.

16. "These data including the directional effects are concordant with the present findings in human." Based on your tree plot, the deletions have opposing effects to what you calculated in this study. So does this mean the lifespan variants are associated with *increased* expression of both?

17. "A disease network showed that cerebrovascular accident and cardiovascular diseases..." Something to the effect of "We generated a disease network using the DisGenet data that showed ..."

18. "We thus leveraged and imputed GWAS summary statistics... to perform MR analyses." I'm confused how you did this. How did you impute summary statistics? I can't find any mention of imputation in the methods. And it's a stretch to impute *summary statistics* anyway without extensive explanation.

19. "When considering associations remaining significant after the Bonferroni correction, 75% of eGene-disorder association pairs were concordant with lifespan." By concordance do you mean increased lifespan was concordant with increased or decreased disease risk?

Discussion

20. "However, the present data suggest... T cells are likely key players involved in the human lifespan." That's a pretty big leap. I might change likely to "me be key players."

21. "Recent data in Caenorhabditis... that invalidation of bec-1..." Should that be inhibition of bec-1? Also in the next sentence. "...the invalidation of bec-1 at the developmental stage..."

22. "Gain of function mutation of PTPN22 was previously..." Is this a mutation? Or is it a gain of function polymorphism? Rare variant? Mutation implies sequence in != sequence out. Unless the reference defines a variant that is de novo in affecteds or exceptionally rare, I would change phrasing.

23. "In a disease network, we observed that cardiometabolic traits and diseases occupied a central position in linking autoimmune disorders and cancer." You don't have to necessarily change anything, but just an FYI this is a tricky thing. Hyperactivation of the immune system may have *direct* consequences for CVD, i.e. the immune disease causes CVD, not vice versa. Additionally, immune surveillance for novel antigens is critical to limit the growth of cancer in vivo.

Methods

24. Genetic association. Did you actually run these associations? Or was it summary stats from the referenced paper (Timmers 2019)? If you used the summary statistics, the method section on how the association was calculated is not necessary. If you did, the provided information is not sufficient. What tools did you use? Or what statistical programming language? If this is from "We thus leveraged and imputed GWAS summary statistics totaling..." there is much information to be filled in. The minimum requirement for methods reporting is sufficient detail that someone could reproduce your analysis if given the same data.

25. GARFIELD – "their significance determined by generalized linear model testing." Is that performed in the command-line c++ code, or is this done in R afterward? And if so, directly implemented by the package or did you do the linear models? Again, not enough detail to reproduce.

26. FUMA – "Chromatin interaction mapping for lifespan... SNP2GENE function using Hi-C data in GM12872..." Is this correct or a typo? All I could find in FUMA was GM12878, though I may have missed it.

27. Single-cell analyses – ASAP lets you perform DE by selecting specific cells or clusters. What were you testing for differential expression? Monocyte vs. DC? Clusters? Other?

28. Publicly available data – information for each GWAS should be included but all supplementary.

Suggestions

General

29. Somewhere in the supplementary material the columns for each supplementary table need to be explicitly defined. I knew most (but not all) of them, but readers shouldn't have to guess.

30. There are many people (myself included) that have an acknowledged personal bias against adding "-ome" onto many things. Connectome isn't *bad*. But you might consider other ways of expression the sentiment.

Introduction

31. "The identification of eQTL genes... key pathways underpinning aging." Perhaps something like "We hypothesized that better understanding of the functional consequences of these regulatory variants might also provide insight into mechanisms of human aging."

32. "In addition, the identification of eGenes... using causal inference." You're dense into MR nomenclature here when you could be giving alternative explanations that are easier to follow. This is critically important at this stage, as the first-time reader may get to parts that are difficult to follow and give up on the paper. Potential alternative:

"In addition, using eQTLs may give us the opportunity to analyze whether the variants are correlated with aging phenotypes or directly causal by using Mendelian Randomization (MR) techniques. The main assumption of MR is that the variables you measure, called instrumental variables (IVs), only affect the outcome through the exposure and without confounders. In other words, if the genetic variants in non-coding RNA only affect aging through their role in altering gene expression (cis and

trans eQTLs), then we can determine the likelihood of their playing a causal role in aging. This strategy of considering independent gene variants in an allelic series as IVs is a powerful tool for causal inference. However, it is not without challenges. If a genetic variant is associated with the output through an alternative mechanism (often referred to as horizontal pleiotropy), it may lead to inflation of type I error. [AN EXAMPLE OF HOW THIS MIGHT WORK]. Different statistical approaches have been developed to assess the robustness of the association discovered by MR and mitigate false-positives. The Cochran's Q test for heterogeneity and Egger regression intercept test are routinely used to detect these associations that may instead be driven by horizontal pleiotropy... By combining these approaches, we can perform a robust estimate of causation for the genetic variants associated with longevity. "

33. "A key advantage of MR... independent IVs for the exposure." Sentence seems out of place here. Might want to move up to directly after you mention MR.

34. "The lifespan is intricately linked with the development of risk factors and disorders." Next paragraph has a similar density to the one above. Possibilities to deconvolute:

"Human lifespan potential is intricately intertwined with the development of disease. As such, aging trajectory and lifespan potential is variable throughout a person's lifespan and may be altered by the different morbidities. We hypothesized that some genetic variants may exhibit antagonist pleiotropy. That is, some variants may provide a survival advantage or reproductive advantage earlier in life, but predispose to disease later in life. Rather than focusing on the individual alleles, we were curious whether similar genetic pathways were involved in chronic disease and lifespan potential. Gene expression is controlled in tissue-specific dynamic networks, with some genes coordinating the activity of specific functional modules. Rather than focusing the shared specific variants between aging and chronic disorders, we instead sought to compare the genetic networks identified for aging and for several chronic human diseases. In this work..."

In both the examples of potential re-writes to the Introduction, take the advice with a grain of salt as far as specifics go, but be aware that readability will increase paper impact.

35. Supplementary Figure 1 is critical to understanding the overall experiment. It may be worth mentioning in the introduction, i.e. "On overview schematic of our study analysis pipeline is presented in Supplementary Figure 1."

Discussion

36. "A model relying... would be canalized for a longer and healthier lifespan (Suppl. Figure 3)." Personally I didn't get much added value from this section or the supplemental figure. Most of this section describes our modern understanding of human genetics: all traits are complex, selection acts on current environment and only seeks to maximize reproductive potential, and environment changes throughout life and exposures. You don't have to alter any of this depending on other reviewer opinions. But my gut reaction would be to either cut it or rewrite it extensively.

Figures

37. All the population allele plots (Supp. Fig 2 and Fig 5b) – you could probably remove all the global allele plots without any reduced understanding of the work.

38. Figure 2 – where b is referred to as left, middle, and right, could instead be an a-d instead of just a, b.

39. Figure 3 – same suggestion. 3e could be split into Fig 3e-g.

40. Figure 6 – might want to use scientific notation, or otherwise use consistent decimal places for P in

panel D.

We thank the reviewers and editor for their constructive comments

Reviewer #1 (Remarks to the Author):

In the present work, the authors have conducted a study showing that genetic regions with a potential influence on lifespan are enriched in open chromatin in blood cells regulating a set of expressed genes involved in immune responses. Using a two-sample Mendelian randomization they then identified blood eGenes associated with the lifespan and a longevity. Overall the present work conveys a rather important message that genetically-determined expression of genes in blood cells is associated with the lifespan and that immune responses play a significant role in the human lifespan trajectory. There are, however, certain concerns that need to be clarified or re-addressed.

1. The work rather fails to answer the very question it poses already in the abstract “How the expression of genes (eGenes) connects the lifespan with the risk of disorders”. The work provides no mechanism or mechanistic ques but correlations and associations. In particular, the work identified an over-representing gene network related to innate immune signaling in blood cells and then moves on making various, occasionally unnecessary, extrapolations. It also remains unclear what the specificity of the immune-related genes is to other tissues.

A: The lifespan is partially genetically determined. So far, the largest published GWAS including more than one million parental lifespans has reported 12 loci (Timmers et al. eLife 2019, PMID: 30642433). This GWAS is overrepresented by noncoding SNPs located in intergenic and intronic regions (please see the Suppl. Figure 2 showing the enrichment in the noncoding genome for the lifespan GWAS), which is similar for other complex traits/disorders (point highlighted at page 5, lines 16-17 of the revised manuscript). Hence, deciphering the impact of noncoding gene variants may provide key insights as to how the expression of genes is linked to traits and disorders. Studies have consistently highlighted that traits/disorders-associated SNPs were enriched in cis-regulatory regions and eQTLs (Maurano et al. Science 2012, PMID: 22955828). The present goals were to assess tissue enrichment of this GWAS and to map eQTL genes (eGenes) in order to perform Mendelian Randomization (MR). MR is a form of causal inference widely used and studies have highlighted that genes identified in GWAS and MR are more likely to represent target for treatment in different disorders (Floris et al. Trends in Genetics, PMID: 29803319). It is worth highlighting that in the present work we mapped 16 genes causally associated in MR with lifespan and that 14 of these genes were not previously identified. Several of these genes expressed by blood cells with strong eQTLs were not previously identified in basic studies; hence present findings may provide a map for follow-up studies. We used stringent criteria for MR and a series of genomic analyses were conducted to provide further insights about the link between these eGenes with disorders. Noteworthy, several eGenes that were found causally associated with the lifespan were also associated in MR with different disorders as highlighted at page 17, lines 12-15. With regard to tissue specificity of expressed genes involved in traits and disorders, this is an interesting and complex issue. There are several processes involved in disorder-tissue-specificity relationships (for a recent and comprehensive review please see Hekselman et Yeger-Lotem Nature Review Genetics 2020, PMID:

31913361). *One common mechanism involved in complex traits is that noncoding risk variants are enriched in regulatory regions that are active in a 'susceptible' tissue. To this effect, by using GARFIELD we found a very strong enrichment of lifespan GWAS in open chromatin of blood cells. In the same line, we found strong eQTLs (low P values) in the blood that were associated with the lifespan GWAS and MR provided another layer of support. The blood is certainly not the only tissue involved in the lifespan, but current analysis of the largest GWAS support that gene variants are strongly enriched in regulatory regions of blood immune cells. Further work is of course necessary to delineate tissue specificity and complex cross talk between different tissues.*

2. There have been many and important works connecting DNA damage, inflammation and ageing or immune related responses and longevity. It is not clear in what sense the present work is novel or what the new message is that was unknown prior to this work. This should be clearly stated.

A: Thank you for raising this point. We agree that insightful basic studies, many in model organisms, have been performed and have highlighted the contributions of immunity, metabolism, protein folding response and autophagy, to name a few, to the lifespan. These studies provide an invaluable resource to understand the ageing process. However, to our knowledge, the present work is the first to report systematically in human the genes genetically expressed associated with the lifespan and to perform MR on mapped eGenes. Using stringent criteria, we report 14 novel genes not previously mapped in GWAS and strongly associated with the lifespan through the expression. Of interest, 2 of these genes OCIAD1 and BECN1 were previously investigated in model organisms and the direction of effects were consistent with our findings (Sinha et al. Blood 2019, PMID: 30952670; Wilhelm et al. Genes Dev 2017, PMID: 28882853). These data thus strongly suggest that this approach is solid and could therefore help identifying novel genes for further follow-up studies. We have highlighted this point at page 9, lines 15-18.

Reviewer #2 (Remarks to the Author):

Mathieu and colleagues applied approaches combined Mendelian randomization, eGenes mapping, functional annotation and network analysis and linked lifespan with the immune response and connected lifespan blood eGenes with chronic disorders. Overall, this is a novel and well-designed work. The findings from this work is important. I have a few concerns.

1. Mendelian randomization used SNPs as instrumental variables for specific exposures. By the Mendelian laws, alleles of SNPs segregate and are randomly inherited from parents to offspring. When using genes for MR analysis, one should be cautious to make causal inferences since multiple SNPs in a single gene may have different function and may not all independent of the outcome.

A: Yes, this is a good point and we agree. We have used different inference tools and different strategies including sensitivity analyses and stringent criteria to limit the risk of horizontal pleiotropy. We have implemented the Cochran's Q test, the intercept test in Egger MR and the MR-PRESSO package in order to detect horizontal pleiotropy and invalid instruments. We have added at page 21 and lines 12-14 a limitation section in the discussion.

2. When considering multiple testing, why use 0.05 as cut-off for FDR? How many genes were tested in the model?

A: The analysis for a long-livedness had limited power owing to the small number of subjects ($n=9793$). No eGenes passed the stringent Bonferroni correction, but 5 genes passed the p FDR threshold of 0.05. These points are now in the revised version of the manuscript at page 16, lines 16-17.

3. When evaluating the associations for disorders, it will be interesting to look at the tissue-specific eQTLs and eGenes.

A: Yes, this is a good idea. We agree that the genes identified in MR and tested in the blood and different disorders may also be associated in other tissues. However, including these data for instance in all the GTEX tissues would necessarily imposed a burden on multiple testing and expand the size of the actual paper, which already includes many approaches and analyses. We agree that future work should examine for instance the role of these genes in different tissues.

4. Limitations of the study should be addressed.

A: We agree, as mentioned above, we have inserted a limitation section at page 21, lines 12-16.

Reviewer #3 (Remarks to the Author):

Summary

Cignon et al present a study in which the causal function of longevity associated SNPs is explored using a wide range of existing data. This is an excellent case study on the importance of large public datasets and databases. In this work they discovered a number of genes causally associated (by Mendelian Randomization) with lifespan and long livedness. These causally associated genes are enriched in immune system functions, and their associated pathways are often associated with chronic morbidity. This synthesis raises many interesting questions regarding the role of the immune system in normal aging.

Clarifications & required changes

General

1. The paper is quite dense. Reviewing the paper took much longer than anticipated

because of the number of times I had to re-read to get a sense of the overall work. Throughout the entire paper, it would benefit you to more explicitly connect ideas and goals. Many paragraphs would benefit from the use of topic sentences to orient the reader to what will be discussed.

There are also areas where the paper is jargon-dense. Many (of these ideas can be explained in ways that will be accessible to readers that aren't experts in the specific domain.

A: Yes, the paper contains many analyses by using different approaches. We have integrated the different suggestions of the reviewers in order to ease the reading and to introduce the analyses performed throughout the paper. The manuscript has been considerably modified and suggestions of the reviewers have been included to ease the reading including by a large public.

2. eGenes is variably defined as eQTL genes or expression genes.

A: Yes, we have clearly stated at page 3, lines 10-13.

Introduction

3. "The strongest association... is in the APOE locus..." An underpinning of the study is that we need to look at eQTLs in non-coding RNA to understand lifespan. Of the 12 loci, how many are actually concentrated in non-coding regions? If of 12 loci the lead SNPs are in non-coding areas in 11/12, this makes good sense. If most are in coding regions, it might now.

A: This is a good point. The GWAS on the parental lifespan was enriched in noncoding regions including intergenic and intronic regions (please see response to point of reviewer#1 and the accompanying figure as well in the revised manuscript at page 5, lines 16-17 and a new Suppl. Figure 2). We agree that the APOE locus is not illustrative for the enrichment of signal in the noncoding region. To this effect the cis-regulated causally associated eGenes were mapped to different loci, but not in the 19 q13.32 locus (figure 3c of the manuscript). Hence, we have removed the sentence related to the APOE locus.

Results

4. "Summary statistics of genetic association data..." did you use the total summary statistics, or the summary statistics for the 12 significant loci?

This is also an example of a paragraph that would benefit from a topic sentence. "We were interested in understanding what tissues the lifespan associated loci might be enriched in. One tool to determine these enrichments is GARFIELD, which uses GWAS summary statistics to generate linkage disequilibrium corrected annotations based on data from the Roadmap Epigenomics and Encyclopedia of DNA Elements (ENCODE) projects. GARFIELD analysis of the lifespan GWAS summary statistics found significant open chromatin enrichments..."

A: We have used the full summary statistics data (now detailed at page 5 of the revised manuscript, page 23, lines 3-6 of the method section). We have modified the section with GARFIELD at page 23, line 23 to page 24, lines 1-3.

5. “highest enrichment in open chromatin detected in CD19+ primary cells... and GM12892”. Not all readers are going to be familiar with the GM12892 cell line or what CD19+ cells are. It is probably worth following this up by mentioning what they are.

A: Yes, we have included the cell annotation at page 5, line 23 and page 6, line 1.

6. “Considering the strong enrichment of lifespan genetic association data with blood...” I might suggest altering blood to something more like immune cells, PBMCs, B cell, lymphocytes, etc, as the signal is enriched in blood, but particularly in lymphocytes.

A: Yes, we have modified to immune cells (page 6, line 2).

7. “...we leveraged summary statistics of blood cis-eQTL data derived from 31,684 samples to map genetic cis-regulated eGenes associated with parental lifespan.” to “...associated with parental lifespan using the FUMA tool.” Or something similar.

A: We have modified and included FUMA at page 6, lines 1-4.

8. “As cis-regulation involves chromatin interactions, we analyzed” to something like “...interactions, we also used FUMA to analyze...” I know you describe these things in the methods section. But the use of so many different tools and DBs makes it hard to follow in the results. A little more detail here and there will help the flow of the paper.

A: We agree, we have included FUMA at page 6, lines 17-19.

9. “Blood eGenes were enriched in gene ontology (GO) for T cell...” Were these enrichments from GARFIELD, FUMA, or EnrichR?

A: These were from EnrichR, we have detailed at page 6, lines 9-12.

10. “...using chromatin confirmation capture (Hi-C) data in GM12872.” Is this GM12872 or GM12878?

A: Thank you, it is GM12878, we have corrected at pages 6, line 19 and page 25, line 2 (in result and method sections).

11. “Blood eGenes used as seeds generated...” to something like “We extracted eGene co-expression networks in whole blood from the database of tissue and cancer specific biological networks. This resulted in a co-expression network with...” Unless I’ve missed how TCSBN works, these aren’t seeds so much as simple retrieval requests for annotated networks.

A: We have modified accordingly at page 7, lines 9-11.

12. "...several eGenes had elevated central betweenness..." The table doesn't seem to be sorted by betweenness or degree. So how are you defining elevated? Greater than the average for the network? The top n %? Percentile? Is there an obvious cutoff?

A: We have included the top percentile (>99%) presented at page 10, lines 1-4.

13. "...A list of blood eGene-derived... provided in Suppl. Table 13." This doesn't have to be changed, but this is an included column in ST12.

A: Yes, we agree. However, we felt that access to the filtered list could be an advantage for the reader.

14. "Analysis of single-cell RNA sequencing data of monocytes..." Again, no transition. Fill the reader in on your logic. "We then wanted to identify what blood cells most expressed BATF3. Analysis of single cell RNA sequencing data..."

A: We have modify the text at page 8, lines 3-5.

15. Suggest moving the whole part about "... involved in cross-presentation of necrotic..." to somewhere with the "These data suggest that lifespan..." It's a bit out of place where it is there. If you put it after the DC association you could then give that brief tidbit about what the cDCs could be doing.

A: We have removed that statement from the result section.

16. "These data including the directional effects are concordant with the present findings in human." Based on your tree plot, the deletions have opposing effects to what you calculated in this study. So does this mean the lifespan variants are associated with *increased* expression of both?

A: The deletion of bec-1 in C. elegans (adult) increased the lifespan and is concordant with the human findings as we found an inverse relationship between the blood expression of BECN1 and the lifespan. The deletion of OCIAD1 in mice decreased the lifespan. In human, we found that the expression of OCIAD1 in the blood was positively associated with the lifespan. Hence, these are concordant in direction. We have added more detail at page 9, lines 16-19 to clarify this point.

17. "A disease network showed that cerebrovascular accident and cardiovascular diseases..." Something to the effect of "We generated a disease network using the DisGenet data that showed ..."

A: We have modified at page 14, lines 7-10.

18. "We thus leveraged and imputed GWAS summary statistics... to perform MR

analyses.” I’m confused how you did this. How did you impute summary statistics? I can’t find any mention of imputation in the methods. And it’s a stretch to impute *summary statistics* anyway without extensive explanation.

A: We have used the package RAISS (Julienne et al. RAISS: robust and accurate imputation from summary statistics, Bioinformatics 2019, PMID: 31173064), an improved and optimized version of the package developed by Bogdan Pasaniuc and Alkes Price (Pasaniuc et al. Fast and accurate imputation of summary statistics enhances evidence of functional enrichment, Bioinformatics 2014, PMID: 24990607), which has been cited 118 times. Briefly, the analysis relies on Gaussian imputation of summary statistics, which according to the work of Pasaniuc et al. recovers the same signal as HMM-based imputation using individual data with no increase in false positive rate. Empirical evidence on 28 GWAS showed a high correlation (0.9-0.97) between imputed and real data. Further detail about the imputation of summary statistics has been included in the method section. This imputation was only performed for the Broer cohort as the initial imputation did not allow the recovery of instruments to perform MR.

19. “When considering associations remaining significant after the Bonferroni correction, 75% of eGene-disorder association pairs were concordant with lifespan.” By concordance do you mean increased lifespan was concordant with increased or decreased disease risk?

A: We considered the direction of effect concordant if : «e.g. the cis-regulated eGene that decreased the risk of disorder/risk factor increased the lifespan». point highlighted at page 15, lines 7-10.

Discussion

20. “However, the present data suggest... T cells are likely key players involved in the human lifespan.” That’s a pretty big leap. I might change likely to “me be key players.”

A: Yes, we agree. We have changed for «...immune response by T cells may be involved in the human lifespan.» Corrected at page 17, lines 21-22.

21. “Recent data in Caenorhabditis... that invalidation of bec-1...” Should that be inhibition of bec-1? Also in the next sentence. “...the invalidation of bec-1 at the developmental stage...”

A: We have modified by using inhibition; page 18, lines 10-12.

22. “Gain of function mutation of PTPN22 was previously...” Is this a mutation? Or is it a gain of function polymorphism? Rare variant? Mutation implies sequence in != sequence out. Unless the reference defines a variant that is de novo in affecteds or exceptionally rare, I would change phrasing.

A: We have changed for ‘A gain of function variant that change an amino acid (R620W) in PTPN22...’ Modifications at page 15, lines 14-16. This is a gain of function well

documented in the literature (Steck et al. Genes Immun. 2009, PMID: 19956096 ; Elshazli et Settin Immunobiology 2015, PMID: 25963842).

23. “In a disease network, we observed that cardiometabolic traits and diseases occupied a central position in linking autoimmune disorders and cancer.” You don’t have to necessarily change anything, but just an FYI this is a tricky thing. Hyperactivation of the immune system may have *direct* consequences for CVD, i.e. the immune disease causes CVD, not vice versa. Additionally, immune surveillance for novel antigens is critical to limit the growth of cancer in vivo.

A: We agree, we have modified for: ‘In a disease network, we observed that cardiometabolic traits and diseases were linked to autoimmune disorders and cancer.’ Modification at page 20, lines 15-16 and also in the abstract at page 2, lines 16-18.

Methods

24. Genetic association. Did you actually run these associations? Or was it summary stats from the referenced paper (Timmers 2019)? If you used the summary statistics, the method section on how the association was calculated is not necessary. If you did, the provided information is not sufficient. What tools did you use? Or what statistical programming language? If this is from “We thus leveraged and imputed GWAS summary statistics totaling...” there is much information to be filled in. The minimum requirement for methods reporting is sufficient detail that someone could reproduce you analysis if given the same data.

A: We used summary statistics from Timmers et al. As suggested, we have removed information related to calculation and referred to the original paper. Modified at page 23, lines 3-14.

25. GARFIELD – “their significance determined by generalized linear model testing.” Is that performed in the command-line c++ code, or is this done in R afterword? And if so, directly implemented by the package or did you do the linear models? Again, not enough detail to reproduce.

A: The GARFIELD package includes a C++ code for data pre-processing and a R code for fold enrichment, significance testing and visualisation. A shell script is provided to run the package. As the software includes linkage disequilibrium and annotation data, only p-value from GWAS are required. We have detailed this point at page 23, line 23 to page 24 lines 1-3.

26. FUMA – “Chromatin interaction mapping for lifespan... SNP2GENE function using Hi-C data in GM12872...” Is this correct or a typo? All I could find in FUMA was GM12878, though I may have missed it.

A: Sorry, it is GM12878. We have corrected at page 6, line 19.

27. Single-cell analyses – ASAP lets you perform DE by selecting specific cells or

clusters. What were you testing for differential expression? Monocyte vs. DC? Clusters? Other?

A: Differential analysis was performed on the DC cluster. Point specified at page 8, lines 5-9.

28. Publicly available data – information for each GWAS should be included but all supplementary.

A: As suggested, we have transfer this information to a Supplementary Table information.

Suggestions

General

29. Somewhere in the supplementary material the columns for each supplementary table need to be explicitly defined. I knew most (but not all) of them, but readers shouldn't have to guess.

A: Yes, we have revised and annotated all columns and included additional information.

30. There are many people (myself included) that have an acknowledged personal bias against adding “-ome” onto many things. Connectome isn't *bad*. But you might consider other ways of expression the sentiment.

A: We have modified the title for 'Mendelian Randomization, Network and Single Cell Analyses Identify Genetically Regulated Blood Genes with the Human Lifespan and Chronic Diseases'.

Introduction

31. “The identification of eQTL genes... key pathways underpinning aging.” Perhaps something like “We hypothesized that better understanding of the functional consequences of these regulatory variants might also provide insight into mechanisms of human aging.”

A: Yes, we changed for ' We hypothesized that better knowledge of the functional consequences of regulatory variants on gene expression might also provide significant insights into mechanisms of human aging.' Modification at page 3, lines 8-10.

32. “In additional, the identification of eGenes... using causal inference.” You're dense into MR nomenclature here when you could be giving alternative explanations that are easier to follow. This is critically important at this stage, as the first-time reader may get to parts that are difficult to follow and give up on the paper. Potential alternative: “In addition, using eQTLs may give us the opportunity analyze whether the variants are correlated with aging phenotypes or directly causal by using Mendelian Randomization (MR) techniques. The main assumption of MR is that the variables you measure, called instrumental variables (IVs), only affect the outcome through the exposure and without confounders. In other words, if the genetic variants in non-coding RNA only affect aging

through their role in altering gene expression (cis and trans eQTLs), then we can determine the likelihood of their playing a causal role in aging. This strategy of considering independent gene variants in an allelic series as IVs is a powerful tool for causal inference. However, it is not without challenges. If a genetic variant is associated with the output through an alternative mechanism (often referred to as horizontal pleiotropy), it may lead to inflation of type I error. [AN EXAMPLE OF HOW THIS MIGHT WORK]. Different statistical approaches have been developed to assess the robustness of the association discovered by MR and mitigate false-positives. The Cochran's Q test for heterogeneity and Egger regression intercept test are routinely used to detect these associations that may instead be driven by horizontal pleiotropy... By combining these approaches, we can perform a robust estimate of causation for the genetic variants associated with longevity. ”

A: We have modified the text accordingly at page 3, lines 10-23 to page 4 lines 1-5.

33. “A key advantage of MR... independent IVs for the exposure.” Sentence seems out of place here. Might want to move up to directly after you mention MR.

A: We have removed this sentence.

34. “The lifespan is intricately linked with the development of risk factors and disorders.” Next paragraph has a similar density to the one above. Possibilities to deconvolute: “Human lifespan potential is intricately intertwined with the development of disease. As such, aging trajectory and lifespan potential is variable throughout a person’s lifespan and may be altered by the different morbidities. We hypothesized that some genetic variants may exhibit antagonist pleiotropy. That is, some variants may provide a survival advantage or reproductive advantage earlier in life, but predispose to disease later in life. Rather than focusing on the individual alleles, we were curious whether similar genetic pathways were involved in chronic disease and lifespan potential. Gene expression is controlled in tissue-specific dynamic networks, with some genes coordinating the activity of specific functional modules. Rather than focusing the shared specific variants between aging and chronic disorders, we instead sought to compare the genetic networks identified for aging and for several chronic human diseases. In this work...” In both the examples of potential re-writes to the Introduction, take the advice with a grain of salt as far as specifics go, but be aware that readability will increase paper impact.

A: We have modified at page 4, lines 7-17.

35. Supplementary Figure 1 is critical to understanding the overall experiment. It may be worth mentioning in the introduction, i.e. “On overview schematic of our study analysis pipeline is presented in Supplementary Figure 1.”

A: That is a good point, we have included the statement at page 4, lines 17-23 to page 5, lines 1-3.

Discussion

36. “A model relying... would be canalized for a longer and healthier lifespan (Suppl. Figure 3).” Personally I didn’t get much added value from this section or the supplemental figure. Most of this section describes our modern understanding of human genetics: all traits are complex, selection acts on current environment and only seeks to maximize reproductive potential, and environment changes throughout life and exposures. You don’t have to alter any of this depending on other reviewer opinions. But my gut reaction would be to either cut it or rewrite it extensively.

A: We agree, we have removed this section. Modifications at page 21, line 10.

Figures

37. All the population allele plots (Supp. Fig 2 and Fig 5b) – you could probably remove all the global allele plots without any reduced understanding of the work.

A: Yes, we have removed these plots.

38. Figure 2 – where b is referred to as left, middle, and right, could instead be an a-d instead of just a, b.

A: As suggested, we have modified.

39. Figure 3 – same suggestion. 3e could be split into Fig 3e-g.

A: As suggested, we have modified.

40. Figure 6 – might want to use scientific notation, or otherwise use consistent decimal places for P in panel D.

A: As suggested, we have modified for scientific notation.

REVIEWERS' COMMENTS:

Reviewer #1 (Remarks to the Author):

My questions on the novelty and specificity of the findings have not been really addressed. Then again, I should give the authors the benefit of doubt. The analysis performed in this work is of potential interest and may be of value to the readership. As is, I am willing to support the publication of the present work.

Reviewer #2 (Remarks to the Author):

The authors have addressed all my concerns. I have no further comments.

Reviewer #3 (Remarks to the Author):

The authors have extensively rewritten the article in response to reviewer comments. The paper itself is easier to follow at this point and I think makes more sense.

All of my comments and suggestions were appropriately addressed (or alternatively had reasonable explanations for not changing). I consider it appropriate for acceptance.